# Hyperactivated PTP1B phosphatase in parvalbumin neurons alters anterior cingulate inhibitory circuits and induces autism-like behaviors

Li Zhang[1,2,8✉], Zhaohong Qin[1,2,8], Konrad M. Ricke[1,2,3,4], Shelly A. Cruz[1,2], Alexandre F.R. Stewart[3,4,5✉] & Hsiao-Huei Chen [1,2,5,6,7✉]

Individuals with autism spectrum disorder (ASD) have social interaction deficits and difficulty filtering information. Inhibitory interneurons filter information at pyramidal neurons of the anterior cingulate cortex (ACC), an integration hub for higher-order thalamic inputs important for social interaction. Humans with deletions including LMO4, an endogenous inhibitor of PTP1B, display intellectual disabilities and occasionally autism. PV-*Lmo4*KO mice ablate *Lmo4* in PV interneurons and display ASD-like repetitive behaviors and social interaction deficits. Surprisingly, increased PV neuron-mediated peri-somatic feedforward inhibition to the pyramidal neurons causes a compensatory reduction in (somatostatin neuron-mediated) dendritic inhibition. These homeostatic changes increase filtering of mediodorsal-thalamocortical inputs but reduce filtering of cortico-cortical inputs and narrow the range of stimuli ACC pyramidal neurons can distinguish. Simultaneous ablation of PTP1B in PV-*Lmo4*KO neurons prevents these deficits, indicating that PTP1B activation in PV interneurons contributes to ASD-like characteristics and homeostatic maladaptation of inhibitory circuits may contribute to deficient information filtering in ASD.

[1] Ottawa Hospital Research Institute, Neuroscience, Ottawa, Canada. [2] University of Ottawa Brain and Mind Institute, Ottawa, Canada. [3] University of Ottawa Heart Institute, Ottawa, Canada. [4] Biochemistry, Microbiology and Immunology, University of Ottawa, Ottawa, Canada. [5] Centre for Infection, Immunity and Inflammation, University of Ottawa, Ottawa, Canada. [6] Cellular and Molecular Medicine, University of Ottawa, Ottawa, Canada. [7] Medicine, University of Ottawa, Ottawa, Canada. [8] These authors contributed equally: Li Zhang, Zhaohong Qin. ✉email: lizhang5210311@gmail.com; AStewart@ottawaheart.ca; hchen@uottawa.ca

ndividuals with autism spectrum disorder (ASD) have impaired social communication, inflexible and repetitive stereotyped behaviors, and cognitive deficits tied to deficient filtering of task-irrelevant information[1]. Inhibitory GABAergic interneurons are key to filtering information in the cortex. While mutations in >100 genes that affect synapse/network formation have been tied to ASD[2,3], targeting ablation of ASD-risk genes only to fast-spiking parvalbumin (PV) interneurons is sufficient to produce similar ASD-like phenotypes as in mice with global ablation[4]. Mice lacking PV interneurons display ASD-like behaviors[5]. In the prefrontal cortex (PFC) of ASD subjects post-mortem[6] and in some ASD mouse models[7], there is a selective loss of PV, but not other GABAergic interneurons. Prenatal exposure to the anti-epileptic valproic acid (VPA) that increases the risk of ASD ~6–10 fold[8] selectively affects PV interneurons[9]. Thus, altered function of PV interneurons affecting the balance of excitatory and inhibitory (E/I) transmission has emerged as an important mechanism underlying ASD.

The medial prefrontal cortex (mPFC) performs higher cognitive processes, including decision making, attention, working memory, emotional control, and social interaction[10]. The mPFC (particularly, the dorsal anterior cingulate cortex, dACC) makes reciprocal connections with the mediodorsal (MD) thalamus[11]. Imaging studies reveal altered function at the anterior cingulate cortex (ACC) and aberrant connectivity between the MD thalamus and ACC in ASD subjects during task performance[12,13]. Hypofunction or lesions at these two regions reduce social interaction in animals[14–16]. An elegant recent study confirmed the relevance of the ACC to social interaction behaviors; selective ablation of the autism-risk gene Shank3 in ACC pyramidal neurons caused social interaction deficits while restoration of Shank3 only in ACC neurons rescued social deficits in global Shank3 mutant mice[17].

Unlike other regions of the thalamus that receive and relay sensory inputs to the cortex, the MD thalamus is a "higher-order" thalamic nucleus that integrates "already processed" information from the layer 5 cortex and relays it back to the layer 2/3 (L2/3) ACC[11]. MD thalamocortical projections excite L2/3 glutamatergic pyramidal neurons and simultaneously activate GABAergic PV interneurons that synapse onto the peri-somata of L2/3 pyramidal neurons and provide feedforward inhibition (FFI)[18]. The fast-spiking PV-mediated FFI provides a temporal filter to limit the "window of opportunity" during which pyramidal neurons integrate excitatory inputs[19]. In addition to peri-somatic FFI, pyramidal neurons also receive dendritic inhibition from somatostatin (SST) interneurons[20]. Together, these inhibitory inputs enable the recruitment of a population of pyramidal neurons in a progressive manner over a wide dynamic range of afferent input strengths; without them, cortical pyramidal neurons would be recruited in an all-or-none fashion that would limit their ability to respond to afferent inputs of different intensities[21,22]. These mechanisms are crucial for information processing and the execution of complex tasks, including social interaction. However, it remains uncertain to what extent deficits in peri-somatic and/or dendritic inhibition contribute to ASD.

Rare single allele deletions containing LMO4 in humans are related to several cases of intellectual disability and one case of autism (Decipher database, https://decipher.sanger.ac.uk/). LMO4 expression is reduced in lymphoblasts of patients with autism[23] and in human cells that carry a mutation in MeCP2 that causes Rett syndrome[24], another neurodevelopmental disorder with ASD-like behavior deficits. Of note, LMO4 is an endogenous inhibitor of the tyrosine phosphatase PTP1B[25,26] that was recently implicated in Rett syndrome[27]. Inhibition of PTP1B eliminated repetitive behaviors and improved motor function in MeCP2-deficient mice[27], suggesting that a severe neurodevelopmental disorder could be ameliorated by postnatal pharmacological intervention targeting PTP1B.

To determine how unopposed PTP1B function in PV interneurons affects cortical function, we examined local- and long-range circuits of the ACC in mice with PV interneuron-specific deletion of the endogenous PTP1B inhibitor LMO4[25,26]. We found that PV-Lmo4KO mice display ASD-like behaviors and have markedly increased PV interneuron excitability that heightens thalamocortical FFI onto the peri-somata of L2/3 ACC pyramidal neurons. Surprisingly, dendritic inhibitory inputs to these pyramidal neurons from non-PV interneurons was reciprocally reduced. Importantly, simultaneous ablation of PTP1B and Lmo4 in PV interneurons was sufficient to restore these inhibitory synaptic circuits and to prevent ASD-like behaviors.

## Results

**PV- *Lmo4*KO mice display ASD-like behaviors**. *Lmo4* is expressed in the medial and lateral ganglionic eminences that give rise to cortical interneurons, including PV neurons[28]. To avoid the embryonic lethality of global *Lmo4* deletion[29], we selectively knocked out *Lmo4* in PV neurons (PV-*Lmo4*KO). PV-*Lmo4*KO mice display reduced social interaction (Fig. 1a) and repetitive stereotyped behaviors (enhanced grooming, enhanced digging) (Fig. 1b, c). Mutant mice are not anxious and display normal olfactory responses to both social cues and non-social cues (Supplementary Fig. 1), suggesting that impaired social interaction probably did not involve olfactory dysfunction. While social interaction increased the number of c-Fos+ neurons (an index of neuronal activation) in layer 2/3 of the dACC nearly eightfold compared to naive wild-type mice, fewer c-Fos+ neurons were counted in PV-*Lmo4*KO mice (Fig. 1d). Despite these deficits, PV-*Lmo4*KO mice show no loss of PV expression or PV neuron numbers (Supplementary Fig. 2), suggesting that loss of *Lmo4* in PV neurons affects other neuronal properties leading to ASD-like behaviors.

**Increased excitability of PV interneurons**. To determine the functional consequence of *Lmo4* ablation in PV interneurons at the dACC, we measured their electrophysiological properties. AAV9 vectors expressing Cre-dependent mCherry were injected to the dACC of PV-Cre/*Lmo4*WT/WT (WT) or PV-Cre/*Lmo4*flox/flox (PV-*Lmo4*KO) mice to label PV interneurons (Fig. 2a). PV interneurons in L2/3 of the dACC showed a relatively depolarized resting membrane potential (RMP) (Fig. 2b) with increased membrane excitability (Fig. 2c) in PV- *Lmo4*KO mice. Importantly, the latency to the first action potential (AP) was markedly shortened (~1.45 ms) (Fig. 2d–f); many features could account for this change, including reduced membrane conductance and/or reduced $I_A$ currents (e.g., Kv1.4 (Kcna4), Kv4.2 (Kcnd2) or Kv4.3 (Kcnd3)). However, further examination revealed no deficit in $I_A$ currents in PV neurons of KO mice (Supplementary Fig. 3), suggesting that the main deficit lies in the reduced resting membrane conductance which is mainly contributed by the leak conductance. Reduced latency to PV interneuron firing would narrow the temporal window to integrate excitatory inputs at pyramidal neurons. Consistent with reduced resting membrane conductance (Fig. 2g), I-V curves of PV neurons (Fig. 2h, i) revealed loss of two conduction components: a voltage-independent component and a voltage-dependent delayed outward rectifier. From these altered kinetics, two types of conductances are likely altered in PV-*Lmo4*KO mice; the voltage-independent two-pore leak (e.g., Kcnk3) and the voltage-dependent delayed outward rectifying potassium conductances (e.g., Kv1.2 (Kcna2), Kv3.1 (Kcnc1) or Kv3.3 (Kcnc2)). This was reflected in the elevated resting membrane potential (Fig. 2b) and

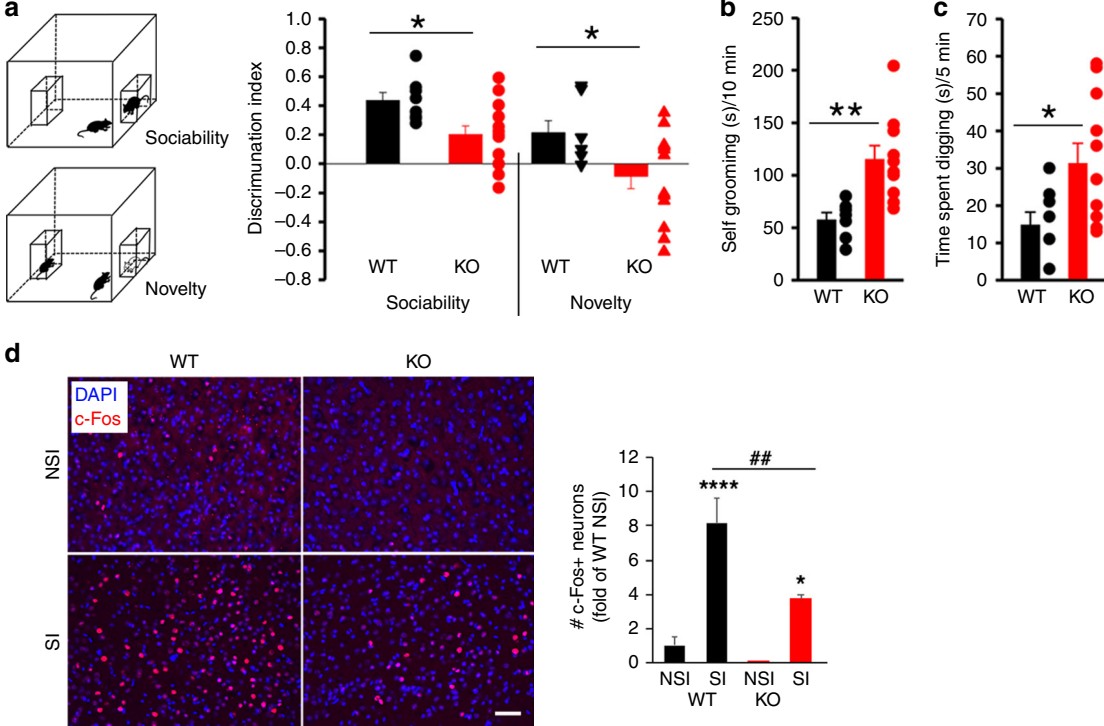

**Fig. 1 PV-*Lmo4*KO mice display autism-like behaviors. a** KO mice have reduced social interaction. The sociability discriminative index compares the time spent in the interaction zone when a littermate is present or absent. The novelty index compares the interaction with a "stranger" to the interaction with a familiar littermate. $n = 8$ WT, 14 KO mice. **b** KO mice show increased self-grooming ($n = 7$ WT, 10 KO) and **c** digging compared to littermate controls ($n = 8$ WT, 11 KO). **d** c-Fos staining revealed neuronal activation in L2/3 of ACC in mice 30 min after social interaction. NSI, no social interaction; SI, social interaction. c-Fos+ neuron numbers were counted and normalized to WT NSI. * and **** are SI compared to NSI for each genotype. $n = 4$ mice per genotype per condition. Scale bar, 50 μm. All data are presented as means ± SEM. *, **, ##, ****, $p < 0.05$, 0.01, 0.01, 0.0001, respectively.

increased width of the AP, with slower decay slope (Supplementary Fig. 3), respectively. Fast after-hyperpolarization (fAHP) was also reduced in mutant PV interneurons (Supplementary Fig. 3), perhaps due to altered function of BK channels. Direct photoactivation of PV interneurons produced less short-term depression of PV-mediated IPSC (Supplementary Fig. 4b, c) implicating a more prolonged and sustained inhibition of L2/3 pyramidal neurons in PV-*Lmo4*KO mice upon repetitive stimulation.

In addition to these altered PV interneuron properties, we also observed increased spontaneous inhibitory but not excitatory synaptic inputs onto L2/3 ACC pyramidal neurons and the E/I ratio was drastically reduced (Fig. 3j, k), with a divisive change in the sIPSC-sEPSC relationship (Fig. 3i). Lack of correlation between sIPSC decay time and rise slope indicates these sIPSCs occur at or near the soma of L2/3 pyramidal neurons, where PV interneurons synapse (Fig. 3b, similar to Supplementary Fig. 4d upon direct photoactivation of PV interneurons).

**Heightened MD thalamocortical FFI**. Given the increased excitability of PV interneurons, we next examined how thalamocortical FFI is affected in PV-*Lmo4*KO mice. MD thalamocortical projections to the dACC were photo-activated after expressing the photosensitive channelrhodopsin ChR2 and mCherry fluorescent marker in these projections with a viral vector (Fig. 4a). Light stimulation activates a modest monosynaptic EPSC and a strong di-synaptic IPSC (from PV interneurons) onto L2/3 pyramidal neurons of the dACC, as described[18] (Fig. 4b). When the light stimulus was adjusted to produce a monosynaptic EPSC in WT and PV-*Lmo4*KO, we

found that FFI was 4× stronger in PV-*Lmo4*KO mice than in controls (Fig. 4b–d), resulting in a reduced E/I ratio (Fig. 4e). The eIPSC-eEPSC amplitude relationship for each individual L2/3 pyramidal neuron showed a divisive change in PV-*Lmo4*KO mice relative to WT, indicating that a larger feedforward inhibitory current occurs with any given excitatory input in PV-*Lmo4*KO mice (Fig. 4c). Of note, the divisive change in the I-E relationship here was similar to what we observed above with the sIPSC-sEPSC amplitude (Fig. 3i). The latency of the di-synaptic IPSC (occurring after the EPSC) recorded from L2/3 pyramidal neurons was much shorter (Fig. 4f), consistent with the faster firing of PV neurons from PV-*Lmo4*KO mice (Fig. 2d). Inhibitory synaptic release probability (Fig. 4g, h) and short-term depression (Fig. 4i, j) were reduced in PV-*Lmo4*KO mice upon repetitive activation of thalamocortical projections; similar synaptic properties were observed with direct photoactivation of PV interneurons (Supplementary Fig. 4b, c), supporting that these PV interneurons mediate MD thalamocortical FFI[18]. In contrast, no difference in excitatory synaptic responses of L2/3 pyramidal neurons to MD thalamocortical activation was observed (Supplementary Fig. 5), suggesting that increased FFI is due to increased PV neuron function rather than altered MD thalamocortical excitatory inputs.

Altered FFI will affect the input–output relationship of pyramidal neurons[21]. Consistent with heightened PV-mediated FFI, L2/3 pyramidal neurons were less responsive to MD thalamocortical inputs (Fig. 4k–p). When 10 sample traces recorded 10 s apart were overlaid for WT and KO upon optogenetic stimulation of MD thalamic inputs, we observed that spike timing was less precise in the KO (Fig. 4k). Spike probability of each pyramidal neuron was measured by the spike

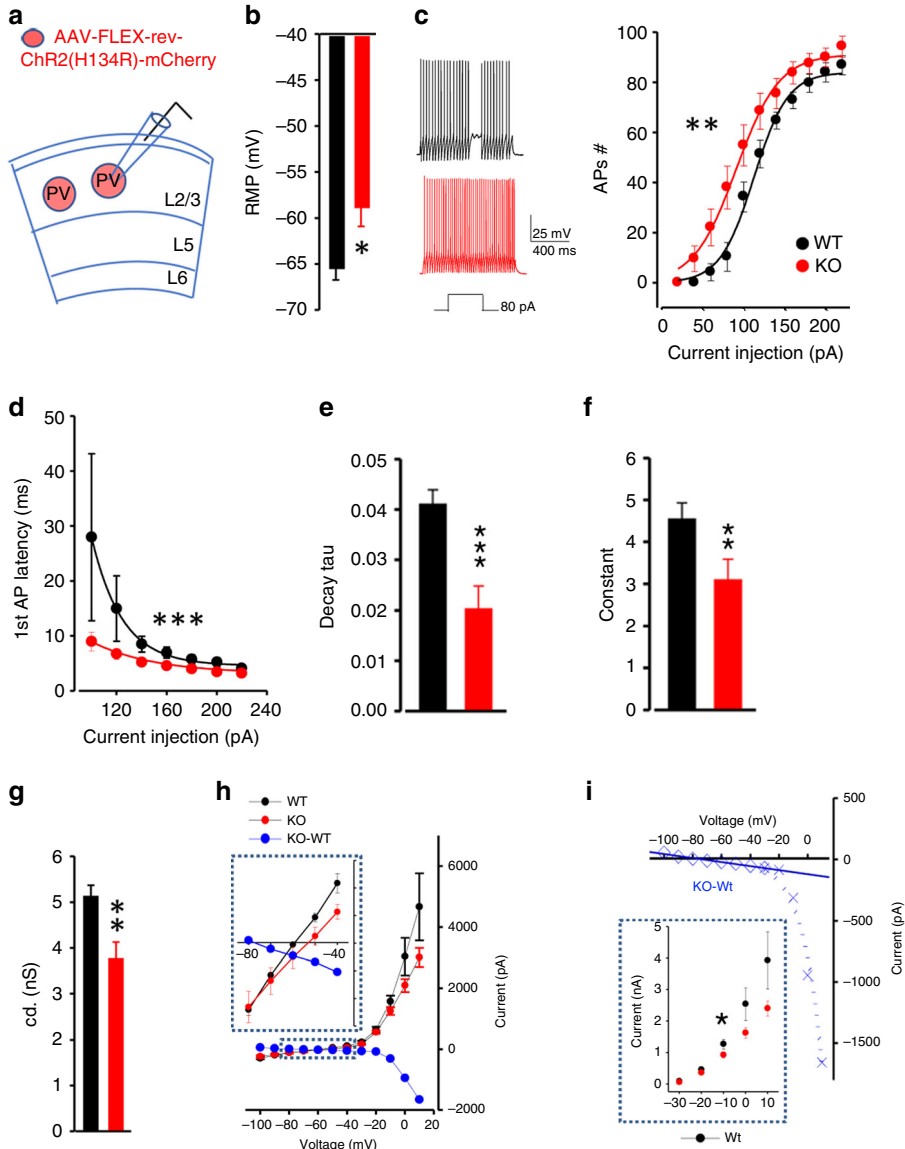

**Fig. 2 Increased excitability of PV neurons in PV-Lmo4KO mice. a** Diagram of AAV9 vectors expressing cre-dependent mCherry stereotactically injected to the dACC of WT (PV-Cre/Lmo4[WT/WT]) or KO (PV-Cre/Lmo4[flox/flox]) mice to label PV neurons. Altered PV neuron properties include: **b** depolarized resting membrane potential, **c** increased membrane excitability (current injection–action potential curve, fitted to a sigmoidal function, shows a leftward shift with reduced threshold and increased number of action potentials). Representative traces are shown. **d–f** Latency to first action potential was reduced ~1.45 ms, $n = 5$ WT, 5 KO mice. Current injection–1st AP latency, fitted to a three-parameter exponential decay, revealed a smaller decay tau (**e**), and a shorter time constant (**f**). **g** Reduced resting membrane conductance. **h, i** I-V curves: the net difference between WT (black) and KO (red) I-V curves is shown in blue. Boxed insets show further expanded current scales to reveal differences. Note (**i**) a linear fit between −100 and −40 mV and an exponential fit above −40 mV revealed reduced voltage-independent (leak) and voltage-dependent delayed rectifying conductances in KO; inset: isolated delayed rectified potassium currents. (Note: all values not adjusted for junction potential of 15 mV). All data are presented as means ± SEM. *, **, ***, $p < 0.05, 0.01, 0.001$, respectively. $N = 9$–11 cells from 5 WT and 8–13 cells from 5 KO mice.

number per each ten photo-stimuli at various strengths until saturated, and fitted to a sigmoid function[21]. With the same EPSC input (measured at Vh = −60 mV, Fig. 4l, inset), spike probabilities of the PV-Lmo4KO cells were suppressed (Fig. 4l, m), by 30% overall (Fig. 4n). The threshold required to induce AP was increased (Fig. 4o) and output gain was much reduced (Fig. 4p). It is intriguing that some WT L2/3 pyramidal neurons responded to MD thalamic activation with a double spike at threshold (Fig. 4m), but this never occurred in PV-Lmo4KO mice, perhaps reflecting the shortened latency of PV-mediated FFI (Fig. 2d).

**Unexpected reduction in dendritic inhibitory inputs to ACC.** L2/3 pyramidal neurons receive at least two types of inhibitory inputs: 1 >peri-somatic inhibitory inputs from L2/3 PV neurons, and 2 >L1 dendritic inhibitory inputs from SST neurons residing in layer 2/3[30] (Fig. 5a). Dendritic inhibition versus proximal (peri-somatic) inhibition affects pyramidal neuron activation differently: dendritic inhibition vetoes dendritic electrogenesis and raises the threshold of pyramidal neuron activation whereas proximal peri-somatic inhibition is important to set the gain (the slope of the output responses relative to input stimuli) and affects how pyramidal neurons integrate

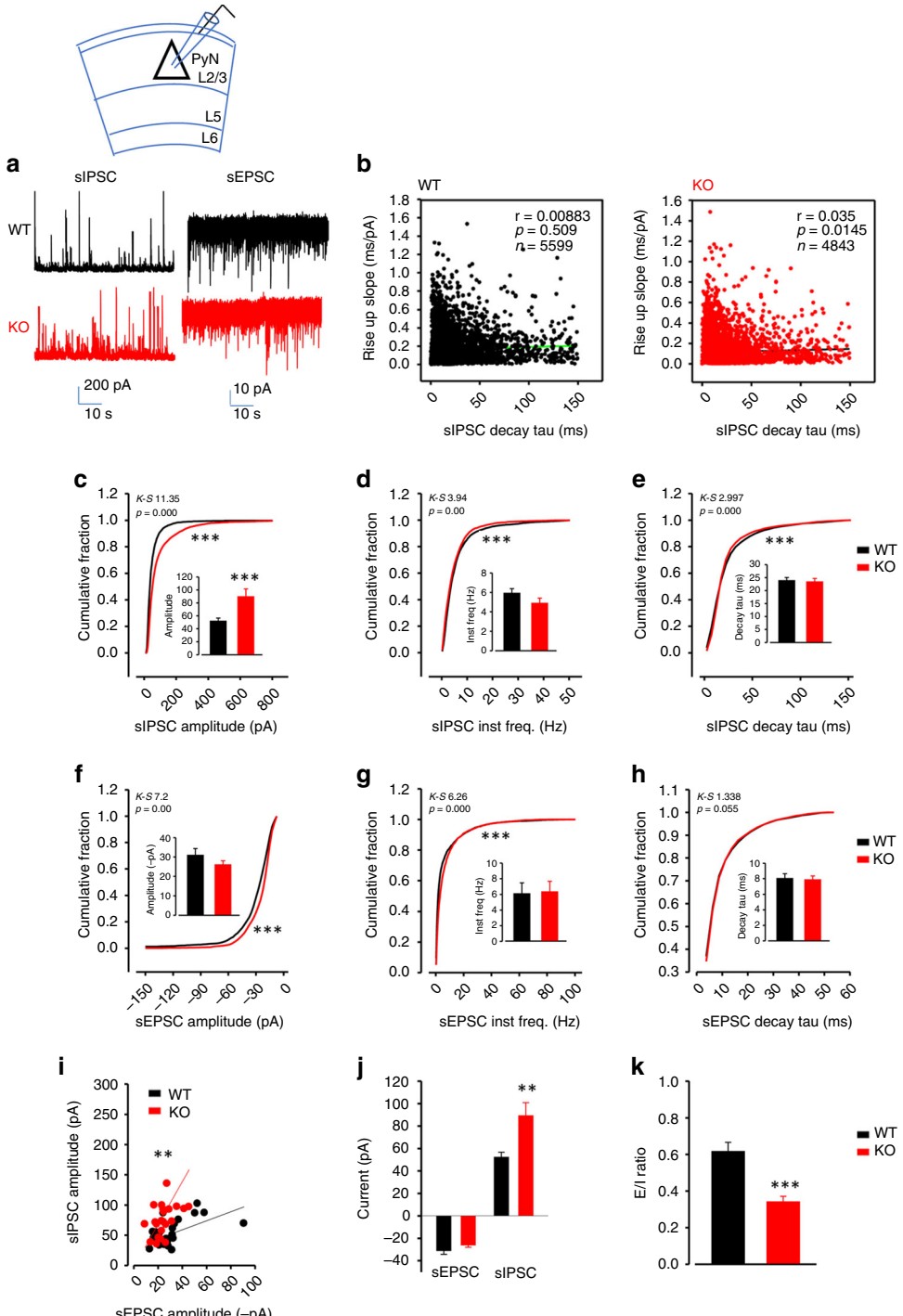

**Fig. 3 Increased spontaneous inhibitory inputs (mostly from PV interneurons) to the somata of the dACC layer 2/3 pyramidal neurons in PV-*Lmo4*KO mice. a** Sample traces of sIPSC and sEPSC of WT (black) and KO (red). **b** Lack of correlation between sIPSC decay time and rise slope indicates these sIPSCs occur at the soma of L2/3 pyramidal neurons, where PV interneurons synapse (compare to Supplementary Fig. 4d). sIPSC (**c–e**) and sEPSC (**f–h**) properties are shown. **i** E-I relationship fitted with linear regression from paired sIPSC and sEPSC amplitudes from each individual cell reveals a divisive change in KO. Increased sIPSC amplitudes in KO without altered sEPSC amplitudes (**j**) resulted in reduced E/I ratio (**k**). $n = 26$ cells/8 WT, 24 cells/8 KO mice. **$p < 0.01$. ***$p < 0.001$.

incoming signals as well as the timing and synchrony of output (spiking)[22,31,32].

ACC L2/3 pyramidal neurons output onto L5 pyramidal neurons that project to MD thalamus as well as to the upper layers of the cortex; the latter cross-laminar projections are thought to be important for inter-laminar feedback integration. As part of this feedback loop, L5 pyramidal neurons activate SST inhibitory interneurons in L2/3 that synapse onto dendrites of L2/3 pyramidal neurons[33–36].

To examine this feedback circuit, electrical stimulation was applied to layer 5. Remarkably, in PV-*Lmo4*KO mice, this produced a smaller FFI onto L2/3 pyramidal neurons with no change in the monosynaptic EPSC (Fig. 5b–d), resulting in a markedly increased E/I ratio (Fig. 5e). There was a subtractive

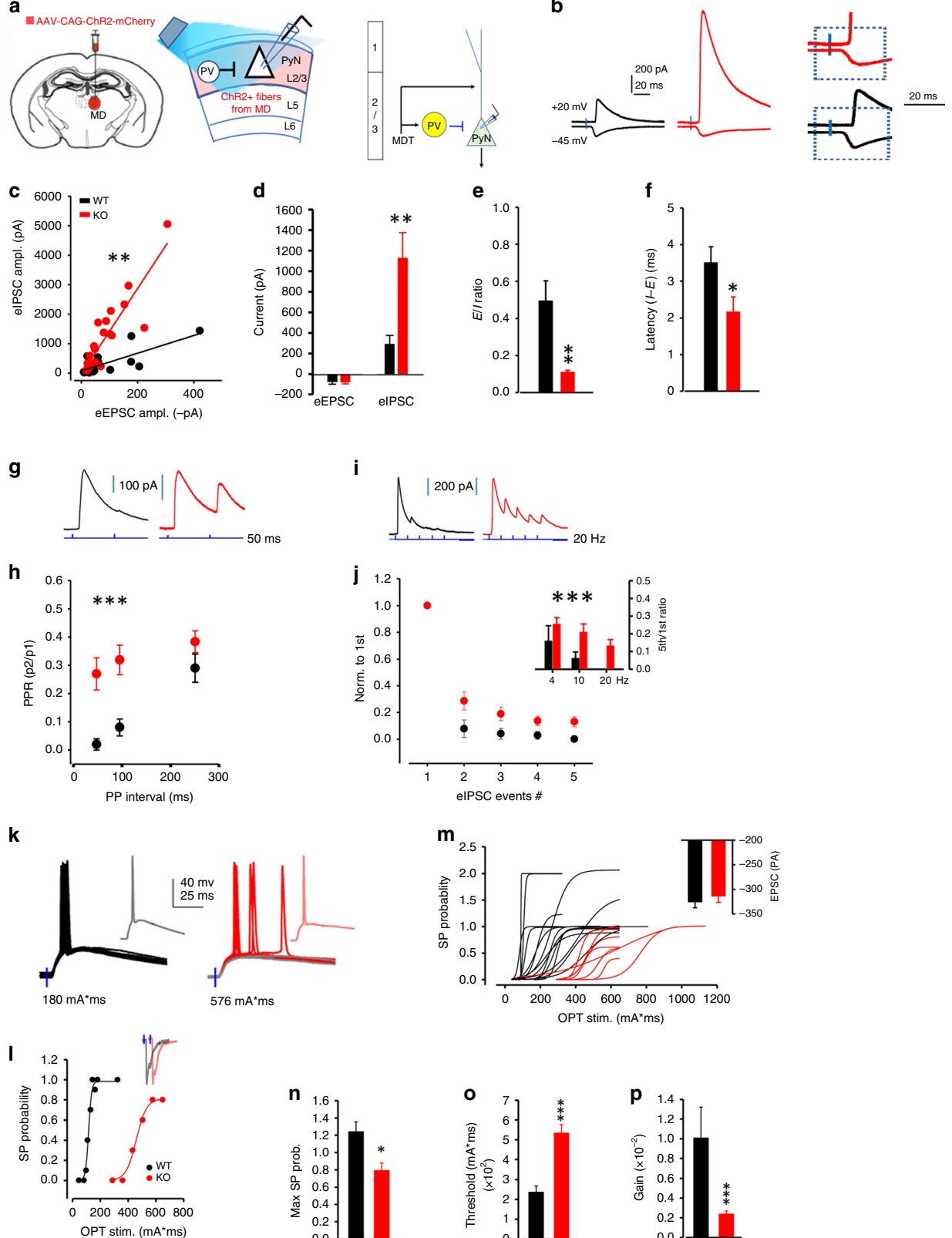

reduction in the I-E relationship between inhibitory and excitatory current amplitudes (Fig. 5c) but no change in the latency of FFI (Fig. 5f). This contrasts to the divisive change in the I-E relationship (Fig. 4c) and shorter latency (Fig. 4f) of PV-mediated inhibition observed upon photoactivation of MD thalamocortical projections. The inhibitory synaptic release

probability and plasticity with layer 5 electrical stimulation also changed in an opposite direction to that observed for PV-mediated inhibition in PV-*Lmo4*KO mice (compare Fig. 5g, h to Fig. 4h, i). Intriguingly, these L5 inhibitory inputs were neither sensitive to DAMGO (a mu-opioid receptor agonist that inhibits GABA synaptic release from PV interneurons and neurogliaform

**Fig. 4 Increased feedforward inhibition in the MD thalamus-dACC circuit in PV-*Lmo4*KO mice. a** Synaptic inputs recorded at dACC layer 2/3 pyramidal neurons (PyNs) after optogenetic activation of thalamocortical projections (left, middle). Right, diagram of MD thalamus-dACC FFI circuit. **b** Sample traces of evoked inhibitory (eIPSC) and excitatory (eEPSC) currents recorded at layer 2/3 pyramidal neurons of wild-type (WT, black) and PV-*Lmo4*KO (KO, red) mice. Light stimulus was adjusted to trigger monosynaptic EPSC in WT and PV-*Lmo4*KO. Traces were enlarged to reveal short latency of monosynaptic EPSC versus longer latency of disynaptic IPSC. **c** E-I relationship fitted with linear regression from paired eIPSC and eEPSC amplitudes from each individual cell. **d** Increased eIPSC amplitude but no difference in eEPSC amplitude, resulted in a reduced (E/I) ratio (**e**). **f** Reduced latency between the rising phase of inhibition and excitatory synaptic responses (I-E) in KO. **g** Sample traces of the eIPSC at a given paired-pulse interval. **h** Paired-pulse ratio. **i** Sample traces of eIPSC in response to repetitive stimuli at 20 Hz reveal short-term depression **j**. Inset, the mean of 5th/1st eIPSC ratio, which represents the degree of short-term depression at various stimulation frequencies. $n = 21$ cells/7 WT, 23 cells/8 KO mice for (**c–j**). **k–p** Under current clamp, L2/3 pyramidal neurons were less responsive to thalamocortical inputs in PV-*Lmo4*KO mice. **k** Sample traces of 10 sweeps, 10 s apart were overlaid for WT and KO. **l** The spike probability from a representative WT or KO L2/3 neuron, plotted against stimulation (light intensity*duration). Insets, representative traces (**l**) and histogram (**m**) show EPSC were of similar amplitudes at holding potential −6 mV with 2 ms light stimulation. **m** Spike probabilities of all WT (15 cells/6 mice) and KO (8 cells/4 mice) L2/3 neurons recorded were sigmoid fitted to obtain (**n**) maximum spike probability, **o** stimulation threshold to elicit 50% peak spike and (**p**) output gain. $n = 15$ cells/6 WT, 8 cells/4 KO mice for (**l–p**). *, **, ***, $p < 0.05, 0.01, 0.001$, respectively.

cells[37]) nor to HU210 (a CB1 receptor agonist that inhibits synaptic release from cholecystokinin interneurons[38]) (Supplementary Fig. 6a) and are likely derived from SST neurons. L2/3 SST neurons receive strong excitatory inputs from L5[34] and provide feedforward inhibition to L2/3 pyramidal neurons in the PFC[36]. We observed a positive correlation between the signal rise (ms/pA) and decay (tau, ms), proportional to the distance traveled, providing evidence that these are (SST-mediated) dendritic inputs, rather than PV-mediated peri-somatic inputs (Supplementary Fig. 6b).

Upon layer 5 stimulation, excitatory inputs to L2/3 showed an increase in paired-pulse ratio and reduced short-term depression for PV-*Lmo4*KO mice (Supplementary Fig. 7), which was different from what we observed for excitatory inputs from the MDT (Supplementary Fig. 5) and may reflect changes in the plasticity of selected local excitatory circuits.

Beyond the feedback loop from L5, L2/3 pyramidal neurons also integrate cortico-cortical and other higher-order thalamic excitatory inputs onto their dendrites in L1. At the same time, these distal dendrites receive inhibitory inputs mediated largely by SST interneurons (and some other local interneurons), but not from PV interneurons[39,40], to provide context-dependent modulation of cortico-cortical excitatory inputs[20]. To determine whether FFI from the cortico-cortical inputs shows a similar homeostatic change as observed in the feedback loop from L5 stimulation (Fig. 5), we carried out electrical stimulation in L1.

Interestingly, in PV-*Lmo4*KO mice, L1 stimulation elicited a reduced FFI response similar to L5 stimulation (compare Supplementary Fig. 8e, f to Fig. 5c, e), suggesting that these (SST) interneurons are also affected by a homeostatic response to *Lmo4* ablation in PV neurons. The reduced dendritic inhibition observed here was surprising since *Lmo4* is ablated in PV but not SST interneurons of PV-*Lmo4*KO mice and suggests a homeostatic compensation in the inhibitory circuits.

**Reduced dendritic inhibition, response threshold, and range.** As a consequence of reduced dendritic FFI, the threshold to activate L2/3 pyramidal neurons in response to layer 1 stimulation (Fig. 6a) was also markedly reduced (Fig. 6b-e). The gain (i.e., the output response relative to input stimuli) of individual neurons was not changed (Fig. 6f), but the population dynamic range (i.e., the range of stimuli that the population of neurons responds to and can distinguish) was much reduced in L2/3 pyramidal neurons of PV-*Lmo4*KO mice (Fig. 6g). At 3.25 V, a stimulus that recruited 92 % of KO cells (22/24) only recruited 30% of WT cells (7/24). The dynamic range to recruit 92% of WT neurons (black dashed line) was twice that of KO neurons.

As reported by Pouille et al.[21], the thresholds and output gains showed no correlation with resting cell membrane potential or

conductance of L2/3 pyramidal neurons (Supplementary Fig. 9), indicating that integration of dendritic inputs rather than the pyramidal neurons' intrinsic properties controls their response threshold and gain.

The leftward shift in the input threshold to recruit pyramidal neurons (without altering their gain) in PV-*Lmo4*KO mice could be reproduced in wild-type mice using low-dose bicuculline (2 μM) to partially inhibit GABAa receptors and was reversed upon washout (Supplementary Fig. 10b–d). Interestingly, if GABA receptors were completely blocked with 10 μM bicuculline, both the threshold and the gain were changed (Supplementary Fig. 11e, f), an effect mediated by APV-sensitive NMDA receptors (Supplementary Fig. 11a, b, e); this is consistent with a previous report that NMDA receptors at distal dendrites increase the output gain[22]. In contrast, upon photo-activation of MD thalamocortical inputs, we did not detect an NMDA component either by voltage clamp (Supplementary Fig. 12a) or current clamp (Supplementary Fig. 12b), indicating that MD thalamo-cortical inputs target the apical trunk of proximal dendrites[33] that lack NMDAR[41], rather than distal dendrites.

**Ablation of PTP1B in PV neurons prevents ASD-like deficits.** Since LMO4 is a dual function protein affecting transcription factors[42] as well as an endogenous inhibitor of tyrosine phosphatase PTP1B[25,26], it was important to determine to what extent PTP1B hyperactivity might contribute to the functional deficits and phenotype that we observed in PV- *Lmo4*KO mice. To this end, we generated PV-DKO (double knock-out) mice that ablate PTP1B together with *Lmo4* in the PV neurons. The social interaction deficits (Fig. 7a) and repetitive behaviors (Fig. 7b) observed in PV-*Lmo4*KO mice were prevented in PV-DKO mice (also see Supplementary Figs. 13, 14). Consistent with rescued behaviors, the number of c-Fos+ neurons after social interaction were not different between WT and PV-DKO mice (Fig. 7c).

PV interneurons of PV-DKO mice showed normalized resting membrane potential (Fig. 7d), excitability (Fig. 7e) and latency to first action potential (Fig. 7f). Membrane conductance (Fig. 7g) was restored, with a normal leak conductance, yet with an exaggerated delayed rectifying current (Fig. 7h, i). The action potential width and fast after hyperpolarization (fAHP) were also normalized (Supplementary Fig. 15). Of note, no difference in basic membrane properties of L2/3 pyramidal neurons was detected (resting membrane potential: WT, −75.4 ± 1.0 (24); KO, −73.2 ± 2.0 (18) and DKO, −72.6 ± 1.0 (24) mV; ANOVA $F(2, 63) = 1.48$, $p = 0.24$; conductance: WT, 3.46 ± 0.27 (24); KO, 3.32 ± 0.39 (24) and DKO, 3.18 ± 0.28 (23) nS. ANOVA $F(2, 68) = 0.19$, $p = 0.83$).

In line with the normalization of most electrophysiological properties of PV interneurons in DKO mice, optogenetic

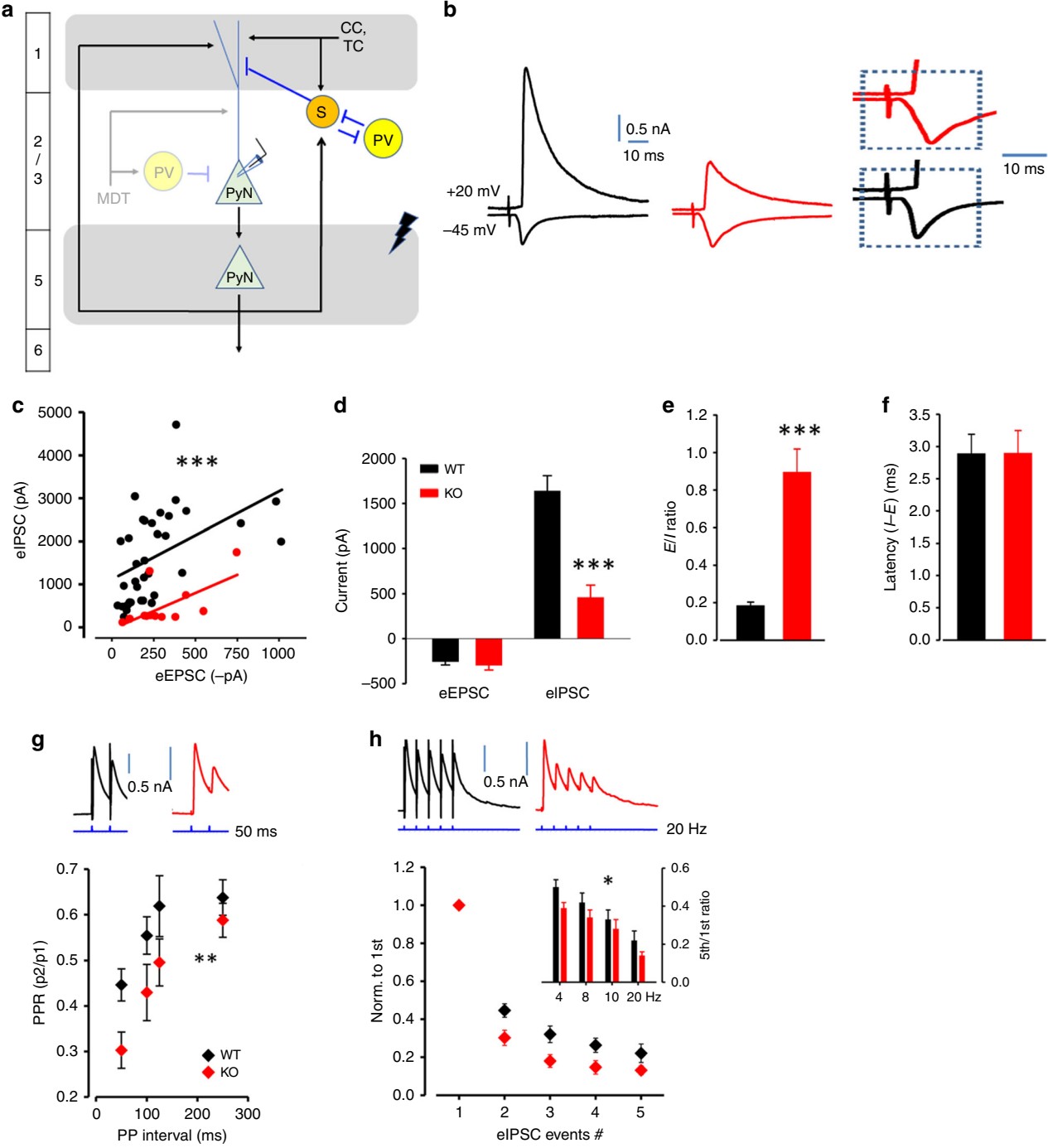

**Fig. 5 Reduced dendritic feedforward inhibition of layer 2/3 dACC pyramidal neurons in response to electrical stimulation at layer 5 in PV-*Lmo4*KO mice. a** Diagram of the inhibitory circuits and placement of stimulation electrodes at layer 5 of the ACC. MDT, inputs from mediodorsal thalamic nucleus onto layer 2/3 pyramidal neurons that project onto pyramidal neurons in layer 5[33]. Layer 5 stimulation activates mainly SST rather than PV interneurons[34]. CC, cortical-cortical inputs. TC, inputs from other higher-order thalamic nuclei to the cortex, for example the ventral medial thalamus[33] and matrix thalamic nuclei[39]. **b** Representative traces of monosynaptic EPSC and di-synaptic feedforward IPSC recorded in L2/3 pyramidal neurons of WT (black, $n = 38$ cells/ 14 mice) and KO (red, $n = 13$ cells/6 mice). Enlarged traces highlight short EPSC and long IPSC latencies, respectively. **c** E-I relationship plotted for individual cells reveals a subtractive shift for PV-*Lmo4*KO mice. Reduced IPSC inputs (**d**) and increased E/I ratio (**e**) in PV-*Lmo4*KO mice. **f** Latency of excitatory and inhibitory inputs to L2/3 pyramidal neurons of WT and KO mice evoked by electrical stimulation at layer 5. In KO, the IPSC showed (**g**) a decrease of PPR and (**h**) enhanced short-term depression. **h** Inset shows the mean of the ratio of 5th/1st IPSC or EPSC at various frequencies. Representative traces are shown above each graph (WT, black; KO red). $n = 13$ cells/5 WT, 11 cells/5 KO mice. *, **, ***, $p < 0.05, 0.01, 0.001$, respectively.

activation of the MD thalamic projections also showed a normalized PV-mediated FFI (Fig. 8a), with restored E/I ratio (Fig. 8b) and latency of inhibition (Fig. 8c). The short-term depression of PV-mediated FFI was also normalized (Supplementary Fig. 16). Furthermore, the stimulation threshold (Fig. 8d, e, g), the maximum spike probability (Fig. 8f), and the output gain (Fig. 8h) of L2/3 neurons to MD thalamocortical stimulation were all restored to WT levels in DKO mice.

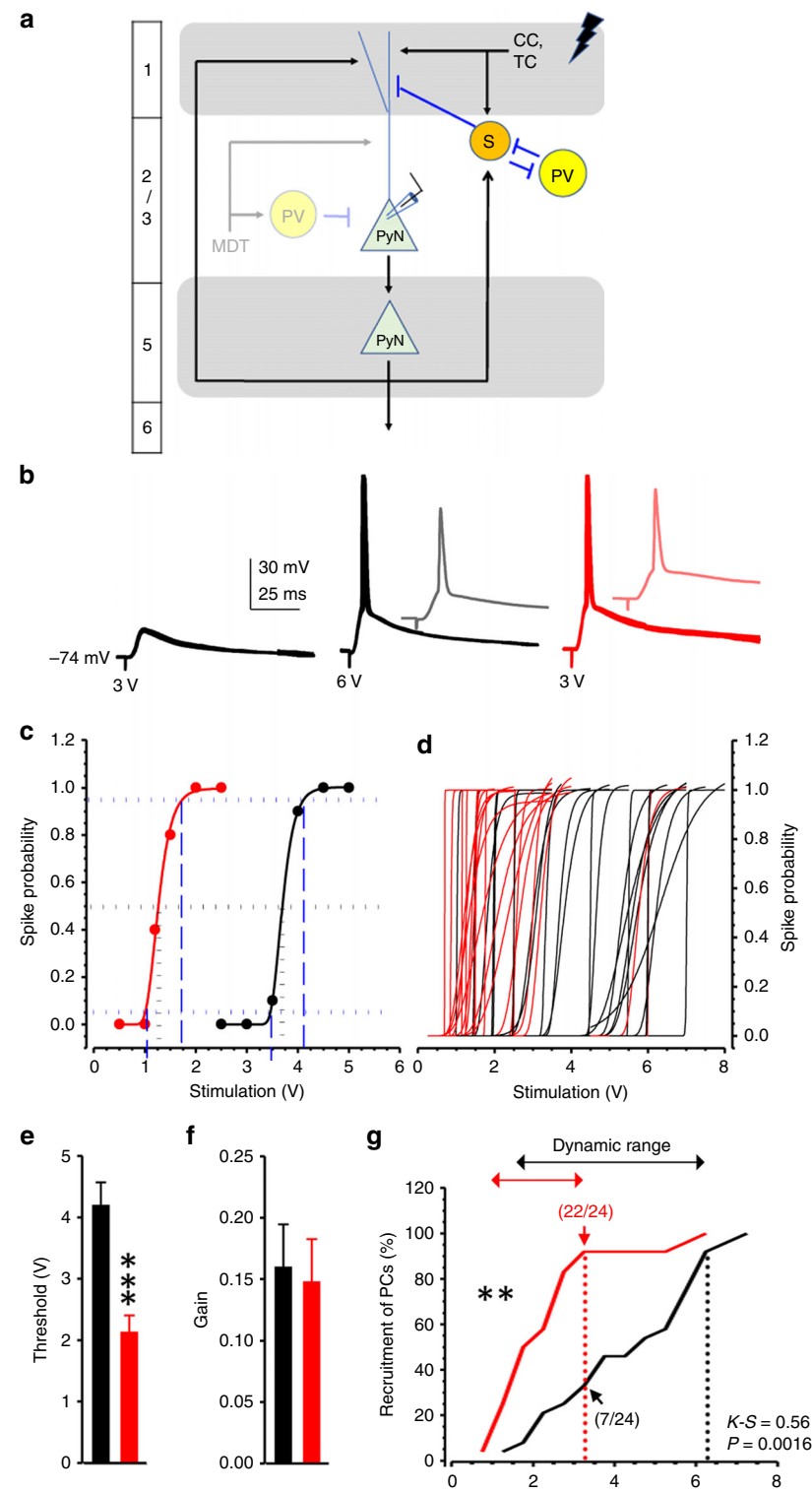

**Fig. 6 Reduced threshold and response dynamic range of layer 2/3 pyramidal neurons at the dACC to electrical stimulation at layer 1, where inhibitory inputs are predominantly from SST but not PV interneurons. a** Diagram of the inhibitory circuits and placement of stimulation electrodes at layer 1 of the ACC. **b** Representative traces recorded at −74 mV showed that an action potential spike could only be elicited in WT neurons (black) by stimulation above 3 V (as shown at 6 V), whereas a 3-V-subthreshold stimulation successfully elicited an action potential spike in PV-*Lmo4*KO neurons (red). **b**–**g** Current clamp recording showed that L2/3 pyramidal neurons in KO mice are more responsive to L1 simulation. **c** The spike probability of a representative L2/3 neuron from WT or KO mice. The horizontal dotted lines mark the 5%, 50% and 95% spike probabilities, and the vertical dashed lines mark the stimulation potentials to reach 5% and 95% spike probability. **d** Spike probability of all WT and KO neurons recorded was sigmoid fit to obtain (**e**) the stimulation threshold and (**f**) the output gain for 50% of maximum stimulus. $n = 24$ cells/8 WT, 24 cells/8 KO for (**c**–**j**). **g** Stimulation thresholds from all cells are cumulatively plotted. Note that a stimulus that recruited 92 % of KO cells (22/24 cells from 8 mice, red dashed line) only recruited 30% of WT cells (7/24 cells from 8 mice). The dynamic range of WT neurons (black dashed line) was twice that of KO neurons. **$p < 0.01$, ***$p < 0.001$.

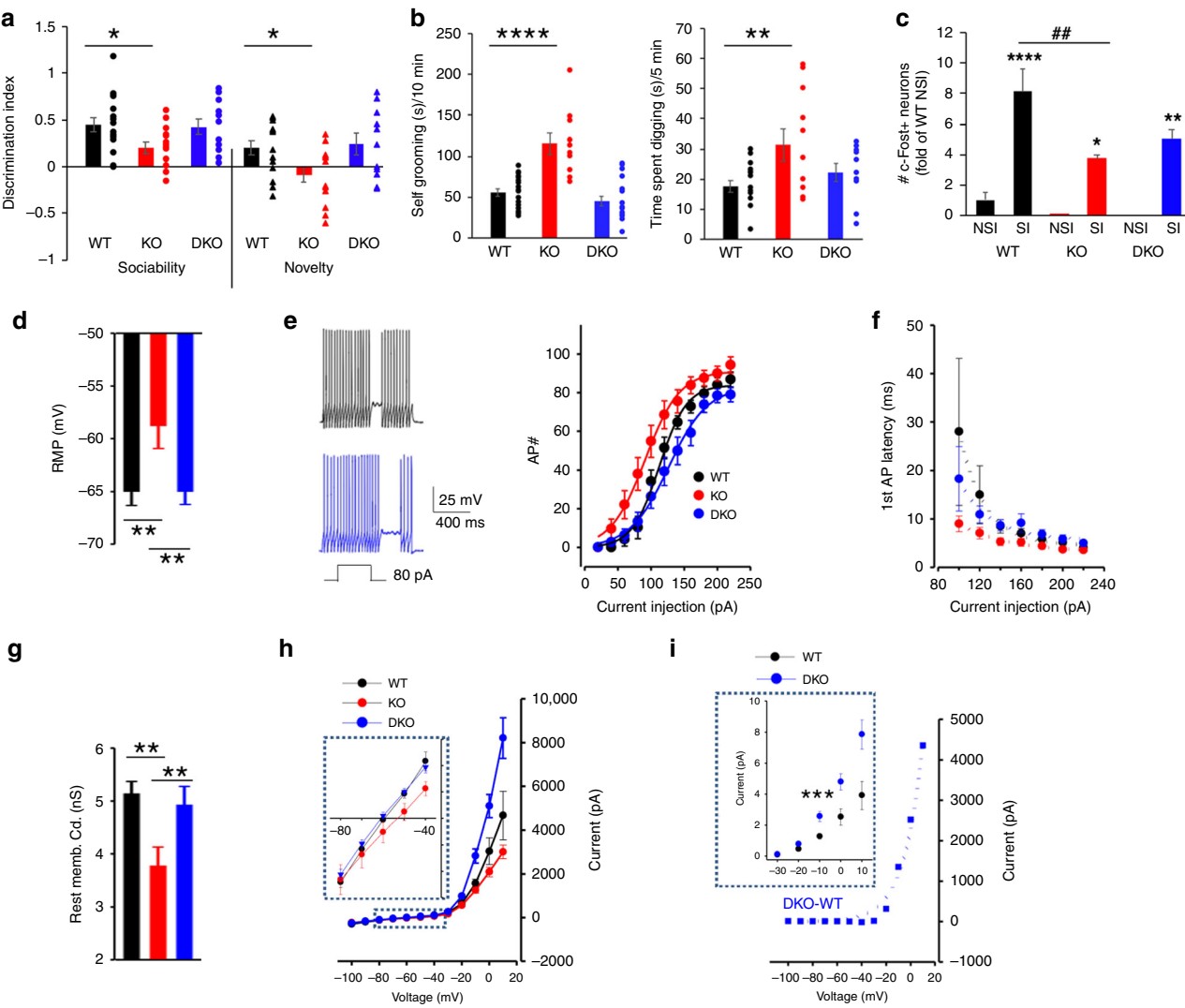

**Fig. 7 Autism-like behaviors and most PV interneuron properties were normalized in PV-DKO mice. a–b** Social interaction and repetitive behaviors. DKO, $n = 11$ (**a**), 17 (grooming) and 11 (digging). **c** c-Fos+ L2/3 ACC neurons in naïve mice and after social interaction. *, **, **** are SI compared to NSI for each genotype. $n = 4$ mice per genotype per condition. In PV-DKO mice, PV neurons showed normalized (**d**) resting membrane potential ($n = 11$ cells/5 WT mice, 20 cells/7 DKO mice), **e** cell membrane excitability plotted as the AP# for a given current injection, fitted to a sigmoidal function, **f** latency for the first spike at various current injections, fitted to exponential decay, and **g** resting membrane conductance. **h** I-V curves of WT (black) and PV-*Lmo4*KO (KO, red) and DKO (blue) revealed normalized leak conductance and augmented (over-compensated) delayed rectifier conductance of PV-DKO mice. **i** The subtraction of I-V curves between DKO and WT is shown and exponentially fitted. Boxed insets (**h**, **i**) show expanded current scales; **i** inset: isolated delayed rectified potassium currents. $n = 10$ cells/5 WT, 12 cells/5 KO, 18 cells/7 DKO for (**e–i**). *, **, ##, ****, $p < 0.05, 0.01, 0.01, 0.0001$, respectively.

Most importantly, SST-mediated FFI of L2/3 pyramidal neurons upon L1 stimulation was also normalized in DKO mice (Fig. 8i-m). Together, these results show that the ASD-like behaviors and electrophysiological changes observed in PV-*Lmo4*KO mice are almost entirely due to disinhibition of PTP1B through loss of *Lmo4*.

## Discussion

Our study shows that selective ablation of *Lmo4* in PV interneurons causes ASD-like behaviors. At the ACC, L2/3 pyramidal neurons receive faster and stronger MD thalamocortical FFI due to increased excitability of PV interneurons and their shortened latency to fire. At the same time, dendritic inhibition of these pyramidal neurons is reduced, lowering their activation threshold without altering their output gain. This compensatory reduction in dendritic inhibition occurs at the expense of the range of stimuli to which L2/3 pyramidal neurons can respond and

distinguish. These changes were all dependent on the selective activation of the tyrosine phosphatase PTP1B, since they were resolved by simultaneous ablation of PTP1B and *Lmo4* in the same PV interneurons in PV-DKO mice.

The most parsimonious explanation is that ablation of the endogenous PTP1B inhibitor *Lmo4*[25,26] activates PTP1B and leads to a myriad of post-translational tyrosine dephosphorylation events. Our previous study reported that ablation of *Lmo4* in glutamatergic neurons (in Camk2αCre/*Lmo4*flox mice) hyperactivates PTP1B that dephosphorylates mGluR5 and impairs mGluR5 function[26]. mGluR5 function may be similarly affected in PV-*Lmo4*KO mice. Altered mGluR5 function has been associated with autism-like behaviors in mice lacking Gprasp2[43]. Moreover, PV neuron-specific ablation of mGluR5 induces social novelty deficits and compulsive repetitive behaviors associated with loss of PV neurons and reduced inhibitory currents (mIPSC) at the hippocampus[44]. In contrast, our PV-*Lmo4*KO mice display social deficits and repetitive

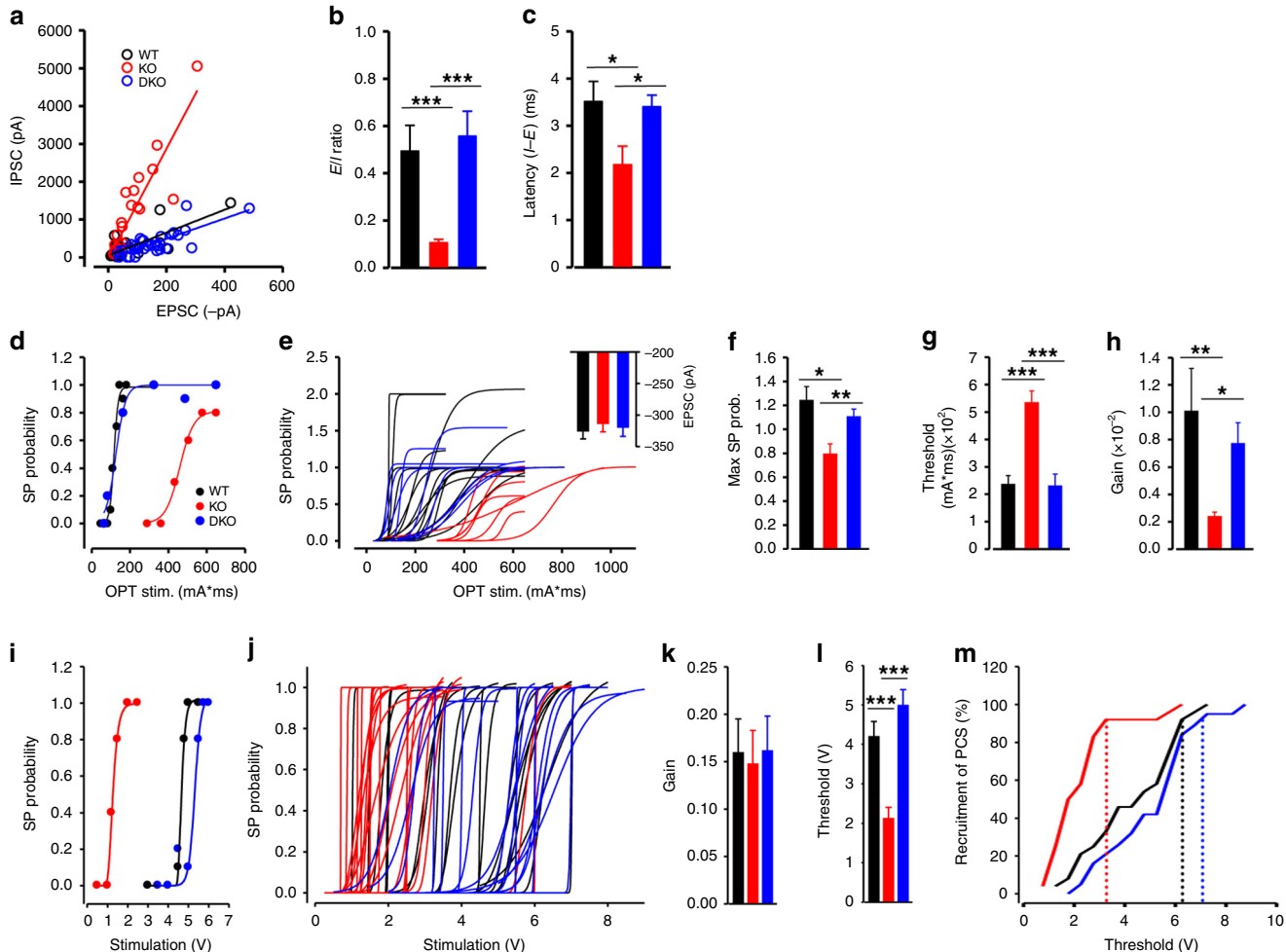

**Fig. 8 The synaptic inputs, response threshold and dynamic range at the layer 2/3 dACC pyramidal neurons are normalized in PV-DKO mice. a–c** MD thalamocortical projections to the dACC were photoactivated to elicit monosynaptic excitatory response at L2/3 pyramidal neurons and compare the di-synaptic feedforward inhibitory inputs. PV-DKO mice showed normalized **a** E-I relationship, **b** E/I ratio, and **c** the latency to the IPSC after EPSC. WT (black), PV-*Lmo4*KO (KO, red), PV-DKO (blue, *n* = 35 cells/12 mice). **d–h** Spike probability (SP) shows restored response of L2/3 pyramidal neurons to MD thalamocortical stimulation in DKO mice (*n* = 9 cells/5 mice). Inset (**e**), histogram shows similar EPSC amplitudes for each of the three genotypes. **f** Maximum spike probability. **g** Activation threshold. **h** Gain. **i–m** Response of L2/3 pyramidal neurons to layer 1 electrical stimulation is restored in DKO mice (*n* = 19 cells/7 mice). Dashed lines show thresholds to recruit 92% of cells for each genotype in **m**. *, **, ***, *p* < 0.05, 0.01, 0.001, respectively.

behaviors without detectable loss of PV neurons. Moreover, both spontaneous and evoked PV-mediated inhibitory currents are increased in L2/3 ACC pyramidal neurons of PV-*Lmo4*KO mice. Although the present study focused on PV neuron function in the ACC, PV neurons in other brain regions including cerebellum[4] and hippocampus[45], could also contribute to ASD-like behavior deficits in PV-*Lmo4*KO mice. Whether compensatory changes in SST-mediated inhibitory inputs are specific to the ACC or also occur in other brain regions is an important question for future studies.

Our previous work indicated that PTP1B activity is normally suppressed by LMO4 in glutamatergic neurons, since knockdown or pharmacological inhibition of PTP1B had no observable effect on behaviors of wild-type mice[26]. Similarly, here we saw no overt effect of PTP1B ablation in PV neurons on social interaction or repetitive behaviors compared to wild-type mice, suggesting that PTP1B function is also normally quiescent in PV neurons. Here, we observed that loss of *Lmo4* in PV interneurons led to a depolarized resting membrane potential and a broadened duration of the action potential likely through reduced leak conductance (*Kcnk3*) and delayed rectifying potassium (Kv1.2 or Kv3) conductance, respectively. Both functions are regulated by

tyrosine phosphorylation[46–48]. These changes in PV-*Lmo4*KO mice were rescued by PTP1B ablation, suggesting that tyrosine phosphorylation-dependent functional deficits in potassium conductances may account for the altered PV function in these mice.

Since PV neuron-specific genetic ablation of PTP1B was sufficient to restore electrophysiological properties of *Lmo4*-deficient PV neurons and to prevent the behavioral and inhibitory circuit deficits in PV-*Lmo4*KO mice, this raises important questions: Might PTP1B hyperactivity in PV interneurons contribute to ASD in humans? Given that the inhibitory circuit is plastic and adaptive changes in adults can occur within days[49], would pharmacological PTP1B inhibition normalize PV neuron properties and restore SST-mediated inhibition?

The di-synaptic nature of fast-spiking PV interneuron-mediated FFI ensures that inhibitory inputs arrive with a short delay and set the window of opportunity during which L2/3 pyramidal neurons can integrate excitatory dendritic inputs and generate action potentials. However, in PV-*Lmo4*KO mice, this window of opportunity is markedly narrowed due to shortened latency of PV interneuron firing coupled with a fourfold heightened PV-mediated inhibition (Fig. 9a). These changes predict

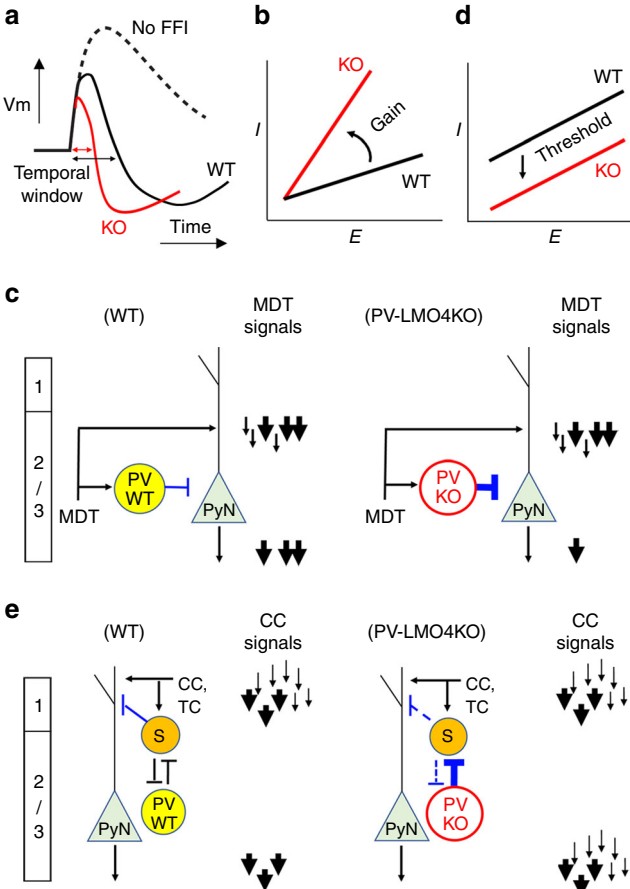

**Fig. 9 Diagram illustrating how homeostatic maladaptation of the inhibitory circuits via feedforward inhibition (FFI) might alter information processing in PV-*Lmo4*KO mice to account for ASD-like behaviors.** This model implicates over-filtering of mediodorsal thalamocortical (MDT) inputs yet inability to filter cortical-cortical (CC) signals. **a** Stronger and faster PV-mediated peri-somatic FFI narrows the temporal window of L2/3 pyramidal neurons to integrate excitatory inputs, resulting in a divisive change in the gain between the excitatory (E) and inhibitory inputs (I) (**b**). **c** Thus, fewer MDT excitatory inputs would succeed in eliciting action potentials at L2/3 pyramidal neurons (PyN); not only background noise (thin black arrows), some valid signals (thick black arrows) would be also filtered away. **d** Adaptive reduction of SST-mediated dendritic inhibition lowers the activation threshold of L2/3 pyramidal neurons, as reflected by a subtractive change between the E-I. This would compromise the ability of the L2/3 pyramidal neuron population to respond/distinguish the range of stimulus intensities from cortical-cortical inputs (CC) or projections from other higher-order thalamic nuclei (TC) (Fig. 6e). **e** In contrast to wild-type mice (PVWT), background noise (thin black arrows) would fail to be filtered in PV-LMO4KO (PVKO) mice.

that MD thalamocortical inputs would be less effective in activating L2/3 pyramidal neurons (Fig. 9b, c). Indeed, we observed a 30% reduction in spike probability (Fig. 4n).

On the other hand, we found an overall reduction in inhibitory inputs from L1 and L5 to distal dendrites of L2/3 pyramidal cells in PV-*Lmo4*KO mice, a feature that should greatly increase pyramidal neuron firing in response to cortico-cortical or other thalamocortical excitatory inputs (Fig. 9d, e). Reduced SST-mediated inhibition would lower the threshold to activate L2/3 pyramidal neurons and background noise would fail to be filtered in PV-*Lmo4*KO mice. Moreover, this would limit the dynamic range of the L2/3 pyramidal neuron population; a stimulus intensity that recruits fewer than one third of L2/3 pyramidal

neurons in wild-type mice would have recruited nearly all of the L2/3 pyramidal neurons in PV-*Lmo4*KO mice (Fig. 6g). The deficit in dendritic inhibition of L2/3 pyramidal neurons was surprising, since *Lmo4* is ablated in PV (but not SST) inter-neurons of PV-*Lmo4*KO mice. Also surprising was the fact that both cortico-cortical FFI in layer 1 and intralaminar FFI from layer 5 to layer 1 were similarly reduced. This could reflect a compensatory change of the local cortical circuit[49] in response to PV interneuron hyperactivity in order to maintain homeostatic inhibition of pyramidal neurons[3]. Alternatively, this could be a hard-wired direct consequence of increased PV-mediated inhibition of SST neurons[50]. This could set up a vicious cycle, since SST neurons tonically inhibit PV neurons[20] and suppressing SST neurons enhances PV-mediated FFI to regulate thalamocortical inputs at the ACC[18]. Whether increased PV-mediated inhibition also causes homeostatic changes in other inhibitory interneurons (e.g., VIP) remains to be determined.

Increased thalamocortical FFI was also reported in a mouse model of ASD induced by fetal exposure to valproic acid[51], where reduced function of *Kcnc1* (Kv3.1) in PV interneurons at the mPFC has been suggested[52]. The notion that altered E/I balance underlies ASD-related deficits may be over-simplified, as both reduced[2] and increased[3] E/I balance have been measured in mouse models of ASD. A recent elegant study of L2/3 pyramidal neurons of the somatosensory cortex in three ASD mouse models by Feldman's group reports that increased E/I ratios are a homeostatic mechanism stabilizing synaptic drive, rather than predicting network hyperexcitability in autism[3].

Our finding of a homeostatic regulation of the inhibitory circuits at the ACC of PV-*Lmo4*KO mice, compartmentalized to peri-somatic versus dendritic inputs, adds a further layer of complexity: a reduced E/I ratio from the MD thalamocortical circuit was accompanied by an increased E/I ratio from the cortico-cortical and other higher-order thalamic inputs. It will be interesting to examine whether compartmentalized changes in the E/I ratio also occur in inhibitory circuits of other brain regions of PV-*Lmo4*KO mice, e.g., in the somatosensory cortex. Another pressing question is whether compartmentalized reciprocal changes in inhibitory circuits occur in other ASD models.

The strength of dendritic and peri-somatic inhibition of cortical pyramidal neurons is dynamically regulated. The compartmentalized switch between dendritic and peri-somatic inhibition offers context-dependent modulation of excitatory activity of pyramidal neurons, increases their flexibility and experience-dependent learning in processing different information inflows[20,22] that are important for social interaction. Activity state-dependent and cortical layer-specific changes in dendritic versus peri-somatic inhibition have also been reported[30,53]. During exploratory behaviors (whisking), L2/3 pyramidal neurons of the somatosensory cortex receive reduced dendritic inhibition from L2/3 SST interneurons but increased peri-somatic inhibition from L2/3 PV interneurons[53]. On the other hand, L4 SST inhibition that targets L4 PV interneurons is increased during whisking[30], and thereby decreases thalamocortical FFI. Unlike the somatosensory cortex, the ACC lacks layer 4 pyramidal neurons and layer 2/3 pyramidal neurons receive direct thalamic inputs. Whether similar changes in inhibition also function at the ACC during exploration and social interaction remain to be determined.

Our c-Fos data revealed that social interaction increased activation of L2/3 pyramidal neurons in wild-type mice but this response was reduced in PV-*Lmo4*KO mice with social interaction deficits. Reduced dendritic (SST-mediated) inhibition onto ACC L2/3 pyramidal neurons observed in PV-*Lmo4*KO mice hints that mutant mice might already be overwhelmed with excessive cortico-cortical and other higher-ordered thalamic

inflows at rest and may be incapable of further increasing attention to (or to distinguish) relevant information during exploration or social interactions. Impaired capacity of neurons to regulate state-dependent synaptic homeostasis may be a major mechanism underlying ASD.

## Methods

**Mice**. PV-Cre/*Lmo4*[flox/flox] (PV-*Lmo4*KO) mice were obtained by mating PV-Cre mice (gift from Dr. Sylvia Arber[1], Jackson Lab # 017320) to *Lmo4*[flox/flox] mice (gift from Dr. Jane Visvader[2]). PV-DKO mice were generated by mating PV-Cre/ *Lmo4*[flox/flox] mice to PTP1B[flox/flox] mice (gift from Dr. Benjamin Neel[3], Jackson Lab #012679). PV-Cre/*Lmo4*[flox/flox] mice were bred with wild-type mice to obtain PV-Cre/*Lmo4*[flox/WT] mice for optogenetic studies using viral vectors. All mice were bred on the C57BL6 background for over 12 generations. Mice were fed with regular chow, and randomly assigned to experimental groups. All procedures for animal use were approved by the University of Ottawa Animal Care and Veterinary Service, and were performed according to institutional guidelines and in accordance with those of the Canadian Council on Animal Care. All studies were conducted using equal numbers of male and female mice.

**Behavioral tests**. Mice ~2–4 months of age were subjected to the following behavior tests modified from ref. [4] with at least 2 days rest in between each test. Investigators were blinded to genotype.

Mice were placed in an open field box (45 cm × 45 cm × 45 cm; MED Associates, St. Albans,VT). Red lighting was used for this test to minimize anxiety form being in a novel environment. The social target mouse (novel or littermate) is contained within a $5.5 × 9.6 × 30$ cm$^3$ (W × L × H) wire mesh cage placed on the opposite side of the square box as described in Fig. 1a. Mice were first habituated for 5 min in the arena with empty mesh cages. Following a 5-min rest in their home cages, they were then videotaped and monitored for 5 min with or without a social target mouse from a mounted camera connected to a computer using Ethovision video tracking software (Noldus). The amount of time they spent at the interactive zone (8 cm away from each side of the mesh cage) was recorded. The discriminative index for sociability was calculated by the formula (A−B)/(A + B), where (A) is the time spent in the interaction zone with a littermate and (B) without any social target in the mesh cage. The discriminative index for novelty was calculated by the formula (A−B)/(A + B), where (A) time spent in the interaction zone with novel strangers and (B) with familiar littermates.

Each mouse was placed in a new cage and videotaped and monitored for 10 min. The time spent licking itself was recorded as self-grooming time.

Mice were placed individually in a new cage with fresh bedding and videotaped. The time spent digging the bedding over 5 min was recorded.

Mice aged 8–12 weeks were habituated in the testing room 30 min prior to testing in their normal home cage. Olfaction was tested as described[4], by measuring the duration of the sniffing response to non-social, then social odors. Odors were presented on cotton-tipped applicators in three consecutive trials per odorant stimulus (2 min per trial) in the following order: water, almond extract, banana extract, social odor 1, social odor 2. Social odors were swipes from cages containing age-matched unfamiliar same sex and opposite sex mice.

**Immunofluorescent staining**. Adult mice were sacrificed and perfused with 4% paraformaldehyde and cryostat sections (20 μm) of various brain regions were subjected to immunofluorescence, as described[5]. Immunofluorescence images were acquired on a Zeiss Z1 fluorescent microscope after Immunostaining using primary antibodies for Parvalbumin (Swant, PV235, ×500), NeuN (Millipore, MAB377, anti-mouse, ×500), and c-Fos (Santa Cruz, sc52, rabbit, ×50), followed by Cy2-, Cy3-conjugated secondary antibodies (Jackson Labs) at ×1000 dilution. For immunofluorescence images, three independent fields at ×20 magnification from six sections were imaged and PV+ cells counted using ImageJ. Investigators were blinded to genotype.

**Stereotaxic surgery**. Two-month-old mice were subjected to unilateral viral injections. To label mediodorsal thalamocortical projections, 0.3–0.35 μl of AAV vectors (AAV9.CAG.hChR2(H134R)-mCherry.WPRE.SV40; Addgene Cat No 100054-AAV9, titer: $2.92 × 10^{13}$ GC/ml) were injected to the mediodorsal thalamus. To label PV interneurons at the dACC, 0.3 μl of AAV vector (AAV-FLEX-rev-ChR2(H134R)-mCherry, Addgene Cat No 18916-AAV9, titer: $1 × 10^{13}$ GC/ml) were injected to the dACC. The following stereotaxic coordinates were used: mediodorsal thalamus, −1.58 mm from Bregma, 0.44 mm lateral from midline, and 3.20 mm vertical from cortical surface; dorsal ACC, 1.42 mm from Bregma, 0.35 mm lateral from midline, and 2.3 mm vertical from cortical surface. To ensure minimal leak into surrounding brain areas, injection pipettes remained in the brain for 5 min after injection before being slowly withdrawn. Electrophysiological studies were carried 2–3 weeks later to allow transgene expression from these viral vectors.

**Electrophysiology**. Mice were anesthetized with isoflurane and decapitated, whereupon brains were quickly removed and immersed in ice-cold ACSF (mM): 124 NaCl, 26 NaHCO$_3$, 10 glucose, 3 KCl, 2.4 CaCl$_2$, 1.3 MgCl$_2$, 1.25 NaH$_2$PO$_4$, equilibrated with 95% O$_2$ and 5% CO$_2$. Brain coronal sections (300 μm) containing ACC were prepared using a microtome (Leica VT1000S) from 2 to 4-month-old mice, as described previously[6].

After >40 min recovery time at room temperature, slices were transferred to a chamber perfused with ACSF 2 ml/min at room temperature. For experiments to test MD thalamic-ACC inputs, slices containing MD thalamus and dACC from each mouse were cut at a 300-μm thickness and imaged to examine the location and extent of ChR2 expression in the MD. Mice were excluded if the extent of infection was too large and had leaked into surrounding brain regions. Rodent MD thalamus lacks interneurons; therefore, all ChR2-infected neurons are expected to be relay projection neurons[7]. For PV-IN, recorded neurons were visually identified using immunofluorescence and infrared differential interference contrast with video microscopy.

In ACC, layers were defined by distance from the pial surface: L2/3: 200–350 μm; L5: 400–550 μm; we recorded our L2/3 pyramidal neurons between 250 and 350 μm. For L5 stimulation, electrode was placed in the middle of L5 which is 450–500 μm away from pia[8]. For L1 stimulation, the electrode was placed on the outer layer of pia. For both L1 or L5 stimulation, the pulse was limited to 0.1 ms with various strengths.

During voltage clamp to obtain spontaneous EPSC/IPSC or evoke EPSC and IPSC to calculate the E/I ratio, recording electrodes were filled with a solution of (in mM): 130 CsMeSO$_3$, 10 KCl, 10 HEPES, 1 EGTA, 2 MgCl$_2$, 10 Na$^+$-phosphocreatine, 4 Na$^+$ - ATP, 0.3 mM GTP (pH 7.3 adjusted with CsOH, 295–300 mOsM); junction potential: 14 mV. In our preparation, the reversal potential for eIPSC was ~ −45 mV measured under NBQX plus D-APV, so we clamped cells at −45 mV for sEPSC/eEPSC. The reversal potential of EPSC was measured at +20 mV in the presence of bicuculline or picrotoxin, so the cells were held at +20 mV to record sIPSC/eIPSC. During current clamp for spike probability, the 130 mM CsMeSO$_3$ was replaced with 137 mM K$^+$-gluconate and 10 mM KCl was reduced to 3 mM in the recording solution (the Cl$^-$ reversal potential was ~ −74.4 mV); junction potential: 15 mV. Unless specified, all potentials mentioned were not adjusted for junction potentials. To ensure slice viability and input continuity between measurements of individual neurons, voltage clamp experiments were carried out in the same cells after current clamp protocols. This approach was also used to obtain basic properties of PV interneurons and L2/3 pyramidal neurons.

Series resistance was compensated at ~50% and monitored during the process, cells with series resistances larger than 25 MΩ were excluded. To characterize PV interneuron and pyramidal neuron membrane properties, we measured resting membrane potential immediately after the rupture of the neuronal membrane. Resting membrane conductance was measured by linear fit I-V curve under voltage clamp between −70 and −40 mV with holding at −50 mV.

To compare fast-inactivating A-type (I$_A$) potassium conductance, TTX (1 μM), Cd$^{2+}$ (100 μM), TEA 20 mM, 4-AP (80 μM) and Ba$^{2+}$ (100 μM) were included in the recording bath solution and currents were evoked in response to 10 mV membrane potential steps from −80 to +40 mV with holding membrane potentials at −80 and −40 mV respectively. The difference between the two currents at these two holding potentials revealed the fast-inactivating A-type (I$_A$) potassium conductance. The delayed rectifier conductance (K$_{DR}$) component from each single cell was isolated by subtracting the voltage-independent component (extrapolated from linear fit between −70 and −40 mV) from the outwardly rectifying currents at each step potential after −40 mV.

Cell membrane excitability was measured under current clamp beginning from −60 mV and the number of action potentials evoked by a series of current injections was determined for all cells. Action potentials elicited by a 140 pA 500 ms square pulse were pooled obtain parameters such as half width, decay slope and fAHP for distribution comparisons and the first ten spikes for real value comparisons (Supplementary Figs. 2, 15). For spontaneous events, 150–200 events recorded from each individual cell were analyzed for amplitude, instant frequency and decay tau analysis; the average values of the 150–200 events from each individual cell were pooled together and compared between groups. The duration of light stimulation for feed-forward inhibition experiments was limited to no more than 3 ms at various intensities.

Spike probability of each pyramidal neuron was measured by the spike number per each ten photo-stimuli or electrical inputs at various strengths until saturated. Data for each single cell or group of cells (membrane excitability and spike probability measurements) were plotted to a sigmoid function to obtain slope or gain and threshold for 50% maximum APs probability, as described[9]. In our experiments, the threshold refers to the electrical stimulation or photoactivation input strength to reach 50% of the maximum response.

To evoke synaptic transmission by activating ChR2, blue light pulses were delivered from a single wavelength LED system (λ = 465 nm; Plexon) triggered by a TTL (transistor-transistor logic) signal from the Clampex software (Molecular Devices). A Multiclamp 700B amplifier connected to a Digidata 1440 interface (Molecular Devices) was used for whole cell recordings and analyzed using pClampfit 10.1. Recordings were filtered at 3 kHz, digitized at 10 kHz.

To compare spike probabilities in response to MD thalamocortical inputs and to test how FFI affects input–output relationships in WT, KO and DKO mice, every cell was held at $-60$ mV to measure EPSC (close to the Cl$^-$ reversal potential, ~ $-74.4$ mV, after adjusting for the 15 mV junction potential) with a 2 ms maximum blue light strength (325 mA) before obtaining spike probability. To ensure cells were given similar input strengths, only cells responding to a photo-stimulus with a monosynaptic EPSC between $-300$ and $-400$ pA were used for analysis.

Where spike probability was measured following L1 or optogenetic MD fiber stimulation under current clamp, cells were subsequently subjected to voltage clamp and held at $-40$ mV to measure an EPSC – IPSC sequence and ensure FFI was triggered by the stimulation or input range (Supplementary Fig. 8). The relative amplitudes of "EPSC" and "IPSC" were analyzed.

The following chemicals and their concentrations were used for recording: D-2-amino-5-phosphonopentanoate (D-APV, 50 μM), 2, 3-dihydroxy-6-nitro-7-sulfamoyl-benzo[f]quinoxaline-2,3-dione (NBQX, 5 μM), (6aR,10aR)- 9-(Hydroxymethyl)- 6,6-dimethyl- 3-(2-methyloctan-2-yl)- 6a,7,10,10a-tetrahydrobenzo [c]chromen-1-ol (HU210, 500 nM), bicuculline (2-10 μM), or picrotoxin (100 μM), and μ-opioid receptor agonist DAMGO (4-10 μM).

**Statistical analysis**. All results are presented as means ± SEM. For between two group comparisons, a *t*-test or one- way ANOVA was used to compare normally distributed data. Data that were not normally distributed as determined by the Shapiro-Wilk normality test were compared with Mann–Whitney Rank Sum test; Z-tests were used to compare parameters obtained from sigmoid and exponential decay functions. For three groups or more, one-way ANOVA or ANOVA on ranks (if normality not passed) was applied, and Bonferroni correction was applied for multiple pairwise testing using two-tailed Student's *t* test for post hoc analysis; for paired-pulse ratio (PPR), short-term depression (STD) and action potential parameters comparisons, two-factor ANOVA compared the main effects of genotype and two-factor interactions; repeat ANOVA was also used for STD analysis. All tests were carried out with SPSS statistical software and differences in means were considered significant at $p < 0.05$. See Supplementary Tables 1, 2 for summaries of statistical analyses.

**Reporting summary**. Further information on research design is available in the Nature Research Reporting Summary linked to this article.

## Data availability
The source data underlying all data Figures and Supplementary Figures are provided as a Source Data file.

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

## Acknowledgements

This research was supported by operating grants from the Canadian Institutes of Health Research (H-HC), the Ontario Mental Health Institute (H-HC), the Canadian Diabetes Association (AFRS), the Heart and Stroke Foundation of Canada (HHC, AFRS), and the Natural Sciences and Engineering Research Council of Canada (HHC, AFRS). We thank Daniel E. Feldman (UC Berkeley) for reading the manuscript and his many helpful suggestions.

## Author contributions

L.Z., H.-H.C., and A.F.R.S. designed the study and wrote the manuscript. L.Z., Z.Q., K.R., and S.A.C. performed the experiments and analysis.

## Competing interests

The authors declare no competing interests.
