## [Peer Review File · Nature Communications]

Reviewers' comments:

Reviewer #1 (Remarks to the Author):

This manuscript reports the role of LMO4 in PV neurons in the regulation of the feedforward inhibition of the thalamocortical pathway onto layer II/III pyramidal neurons in the dorsal ACC. In support of the conclusions, the authors demonstrate that PV neurons lacking LMO4 expression show increased neuronal excitability and increased feedforward inhibition onto GABAergic neurons in L2/3 dACC. Interestingly, this change in perisomatic inhibition is associated with decreased dendritic inhibition mediated SOM neurons, suggestive of adaptive changes. Moreover, these changes lower response threshold and dynamic range of L2/3 pyramidal neurons, likely narrowing down the time window during which L2/3 pyramidal neurons integrate incoming excitatory synaptic inputs. Interestingly, the deletion of the PTP1B tyrosine phosphatase additional to LMO4 rescues nearly all of the electrophysiological and behavioral phenotypes, suggesting that PTP1B is the main mediator of the LMO4-related phenotypes.

The manuscript contains an impressive array of electrophysiological datasets encompassing several cortical circuits including the thalamocortical circuit and associated PV-mediated feedforward inhibition and SOM-mediated dendritic inhibition. In addition, the genetic double KO of PTP1B and LMO4 demonstrates strongly that PTP1B hyperactivation mediates the effects of LMO4 deletion.

Major comments:

1. The authors need to show whether the PV-specific deletion of LMO4 affects the basal behavioral characteristics of the mice, including locomotor activity, anxiety-like behaviors and olfactory function to see if these factors may potentially contribute to the social deficits observed in the LMO4 cKO mice. Ideally, the impacts of the double KO on these behaviors should also be tested.
2. Figure 1a. An ideal setup for social interaction should be the comparison between social target vs. inanimate object because an empty cage and a cage with a mouse is not a fair method to assess social preference.
3. Figure 6a-c. The data from single LMO4 KO should be compared with the WT and DKO results. In addition, does the single KO of PTP1B induce social and repetitive behavioral deficits?

Minor comments:

1. Social interaction and novel recognition in Figure 1 show only the social/novelty preference index. However, the actual time spent in social/empty targets should be presented because it would make it clear which between social and inanimate targets contribute more strongly to the social impairments and rescues.
2. Neither LMO4 nor PTP1B is listed as ASD-related genes in the SFARI database. I guess that the clinical evidence supporting the association of these genes with ASD is weak, and this should be clarified and ASD-related remarks should be toned down.

Reviewer #2 (Remarks to the Author):

The manuscript "Unleashed tyrosine phosphatase PTP1B activity in parvalbumin neurons alters homeostasis of anterior cingulate inhibitory circuits and induces autism-like behaviors in mice" by Li Zhang et al, performs a functional study in the anterior cingulate cortex when LMO4 is conditionally deleted in PV-positive neurons. The authors delve into the alteration inflicted upon cortical microcircuitry and feedforward inhibition to propose deficient information processing as a trait in autism.

The study is comprehensive in terms of circuit dissection, is novel and of interest to the field. The

manuscript is well written, polished and the statistics used are appropriate.

This reviewer is very positive on this work, but there are a few aspects that could improve the manuscript:

1) There is PV-Cre expression in the cerebellum, hypothalamus, hippocampus as well as several other brain regions and not only in the cortex, which is where the authors focus their functional analysis. Thus, the notion that the understanding how "deficits in peris-somatic and/or dendritic inhibition contribute to ASD" or that that this particular circuit is driving the behavioral alterations is misleading.

This work does provide a tour de force in terms of analytical measures of PV-inhibitory dysfunction, but the results cannot be generalized from observations in a specific circuit. To maintain this claim, rather than a "global" PV deletion of LMO4, the authors would need to provide either a PV-specific deletion of LMO4 in the ACC, or conversely perform a rescue of behavior after reintroducing LMO4 expression only in ACC PV interneurons in the "global" PV:LMO4 mice. To clarify, this reviewer does not require these experiments to be performed if the language is suitably changed throughout the manuscript in order to become more tempered and circumspect.

2) A major point conveyed in this work is that the hyperexcitability of the PV-interneurons caused by LMO4 deletion interferes with proper cortical microcircuitry. The authors do a very good job looking at several electrophysiological properties of the circuit, assess local and long-range connectivity, but there a clear lack of mechanistic insight into the cause of the hyperexcitability itself. The authors find indications that action potentials and resting membrane potential are changed in the PV neurons but only putatively indicate that this may be due to altered K⁺ currents while never directly addressing this point. It would be important to directly compare measures of K⁺ currents between genotypes that could allow for the observed differences and depart from conjecture.

3) The behavioral characterization performed in this work is cursory. There are, however, two possibilities that could easily improve the authors claims and link to autism, by either looking at the ACC activity in the context of social behaviors, or by performing tests that are more ACC specific. This would be important particularly since the human genetics data linking LMO4 to autism is not very strong. For example, in the database pointed by the authors there is a clearer connection to intellectual disability (6/9 cases) rather than autism (1/9 cases), so linking this circuit to the behavior analysis is critically important.

Are there behavioral tests that more specifically assess functions that are linked to ACC; or alternatively, happens to c-fos activation in the ACC of their animals following a social stimulus?

4) While never directly intersecting with this work, the manuscript would be greatly improved if the authors discuss and contextualize their new finding in light of previous work showing a dysfunction in mGluR5 signaling in LMO4 KO mice. This would strengthen the discussion by offering other potential avenues that could help explain the animal's phenotype and some of the circuit alterations observed. The authors should also acknowledge recent evidences linking autism/intellectual disability to mGluR5 signaling as not to oblivate this competing (but not mutually exclusive) possibility. In this sense, the following works should be cited.

<https://doi.org/10.1016/j.neuron.2015.02.015>

<https://doi.org/10.1038/mp.2015.113>

<https://doi.org/10.1038/s41467-019-09382-9>

5) Please provide a zoomed in of the optogenetic stimulus onset and postsynaptic response that reflect mono versus di-synaptic response in the example in Figure 3 and 4.

6) Murine genes should be indicated in italic

7) The x-axis on figure 2d is cut

8) In the main text, page 4, Supplementary figure 4b is being indicated, whereas 4c should be the one being noted.

Reviewer #3 (Remarks to the Author):

The present manuscript reports alterations in FFI in the aCC following LMO4 and LMO4-PTP1B in PV neurons KO. The electrophysiological studies are well performed and present important data on the inhibitory circuit. The behavior is convincing but see comment below. IT is a well presented set of data. This study has to the body of knowledge regarding FFI and interneuron contribution to ASD phenotype. I nevertheless have the following set of comment:

Figure 2:

a. Please show some of the AP traces as well as whole cell currents.

b. The IV curves in h show no significant difference. So it is unclear how this could account for the significant change in cd. In addition, there was almost no change at the resting membrane potential and thus it does not seem that the mentioned change in conductance accounts for the difference in resting membrane potential. Actually based on the IV curve, there is no difference around -60 or -65 mV so the cells should have the same resting membrane potential. Please clarify.

c. Some of the data presented in supp Fig. 3 should be incorporated in the main figure. These data such as E/I ratio are important.

In the DKO, what is the expected effect of more KV current on cell firing and integration?

Figure 6c: there is a very large spread in the DKO. Did some of the mice present with locomotion issues?

Reviewers' comments:

Reviewer #1 (Remarks to the Author):

This manuscript reports the role of LMO4 in PV neurons in the regulation of the feedforward inhibition of the thalamocortical pathway onto layer II/III pyramidal neurons in the dorsal ACC. In support of the conclusions, the authors demonstrate that PV neurons lacking LMO4 expression show increased neuronal excitability and increased feedforward inhibition onto GABAergic neurons in L2/3 dACC. Interestingly, this change in perisomatic inhibition is associated with decreased dendritic inhibition mediated SOM neurons, suggestive of adaptive changes. Moreover, these changes lower response threshold and dynamic range of L2/3 pyramidal neurons, likely narrowing down the time window during which L2/3 pyramidal neurons integrate incoming excitatory synaptic inputs. Interestingly, the deletion of the PBP1 tyrosine phosphatase additional to LMO4 rescues nearly all of the electrophysiological and behavioral phenotypes, suggesting that PTP1B is the main mediator of the LMO4-related phenotypes.

The manuscript contains an impressive array of electrophysiological datasets encompassing several cortical circuits including the thalamocortical circuit and associated PV-mediated feedforward inhibition and SOM-mediated dendritic inhibition. In addition, the genetic double KO of PTP1B and LMO4 demonstrates strongly that PTP1B hyperactivation mediates the effects of LMO4 deletion.

Major comments:

1. The authors need to show whether the PV-specific deletion of LMO4 affects the basal behavioral characteristics of the mice, including locomotor activity, anxiety-like behaviors and olfactory function to see if these factors may potentially contribute to the social deficits observed in the LMO4 cKO mice. Ideally, the impacts of the double KO on these behaviors should also be tested.

R> We now provide data on locomotor function, anxiety-like behaviors and olfactory function for the PV-*Lmo4*KO mice compared to littermate controls (see new suppl. Fig. 1). We also provide the data for PV-DKO, PV-PTP1BKO as well. (see new suppl. Fig. 14).

PV-LMO4KO mice had normal olfactory responses to social and non-social odours, but displayed social interaction deficits. As reported previously, impaired social interaction does not necessarily involve olfactory dysfunction^{1,2} and ablation of LMO4 in parvalbumin neurons did not produce a measurable change in olfactory function.

2. Figure 1a. An ideal setup for social interaction should be the comparison between social target vs. inanimate object because an empty cage and a cage with a mouse is not a fair method to assess social preference.

R> Our setup was modeled on a protocol presented in Dr. Jacqueline Crawley's well-cited paper³. Importantly, using this setup we could reliably detect social preference and social novelty in each cohort of littermate control mice and these behaviors were defective in PV-*Lmo4*KO mice.

3. Figure 6a-c. The data from single LMO4 KO should be compared with the WT and DKO results. In addition, does the single KO of PTP1B induce social and repetitive behavioral deficits?

R> Fig. 6a-c (now Fig. 7a, b) includes PV-*Lmo4*KO data. PV-PTP1BKO (single KO of PTP1B, aka., PKO) mice behave similarly to wild type mice (see new supplementary Fig. 13 for behavior tests of social interaction, repetitive behaviours, olfactory function, and anxiety).

Minor comments:

1. Social interaction and novel recognition in Figure 1 show only the social/novelty preference index. However, the actual time spent in social/empty targets should be presented because it would make it clear which between social and inanimate targets contribute more strongly to the social impairments and rescues.

R> The actual time spent in social/empty targets for Fig. 1a and 7a is now shown in supplementary Figs 1 and 13, respectively.

2. Neither LMO4 nor PTP1B is listed as ASD-related genes in the SFARI database. I guess that the

clinical evidence supporting the association of these genes with ASD is weak, and this should be clarified and ASD-related remarks should be toned down.

R> We acknowledge that LMO4 is not included SFARI *current* database. While the SFARI gene database is an evolving database to reflect new discoveries, we have toned down our statement and pointed out that “Rare single allele deletions containing LMO4 in humans are related to several cases of intellectual disability and one case of autism.” (page 3, paragraph 2).

Reviewer #2 (Remarks to the Author):

The manuscript “Unleashed tyrosine phosphatase PTP1B activity in parvalbumin neurons alters homeostasis of anterior cingulate inhibitory circuits and induces autism-like behaviors in mice” by Li Zhang et al, performs a functional study in the anterior cingulate cortex when LMO4 is conditionally deleted in PV-positive neurons. The authors delve into the alteration inflicted upon cortical microcircuitry and feedforward inhibition to propose deficient information processing as a trait in autism.

The study is comprehensive in terms of circuit dissection, is novel and of interest to the field. The manuscript is well written, polished and the statistics used are appropriate.

This reviewer is very positive on this work, but there are a few aspects that could improve the manuscript:

R> We thank the reviewer for appreciating our effort.

1) There is PV-Cre expression in the cerebellum, hypothalamus, hippocampus as well as several other brain regions and not only in the cortex, which is where the authors focus their functional analysis. Thus, the notion that the understanding how “deficits in peris-somatic and/or dendritic inhibition contribute to ASD” or that that this particular circuit is driving the behavioral alterations is misleading.

R> We agree that PV neuron dysfunction in other brain regions besides the ACC can contribute to ASD-like behaviors. We have revised our manuscript to acknowledge this point on page 12 of the Discussion as follows: “Although the present study focused on PV neuron function in the ACC, PV neurons in other brain regions including cerebellum⁴ and hippocampus⁴⁵, could also contribute to ASD-like behavior deficits in PV-*Lmo4*KO mice. Whether compensatory changes in SST-mediated inhibitory inputs are specific to the ACC or also occur in other brain regions is an important question for future studies.”

Nonetheless, the experiment requested by Reviewer 2 (see below) provides compelling evidence supporting that the ACC is involved in social interaction (Fig. 1d: we observed a marked increase in the number of c-Fos positive neurons at the ACC after social interaction in wild type mice but significantly fewer c-fos positive cells in PV-*Lmo4*KO mice that display social interaction deficits.

*This work does provide a **tour de force** in terms of analytical measures of PV-inhibitory dysfunction, but the results cannot be generalized from observations in a specific circuit. To maintain this claim, rather than a “global” PV deletion of LMO4, the authors would need to provide either a PV-specific*

deletion of LMO4 in the ACC, or conversely perform a rescue of behavior after reintroducing LMO4 expression only in ACC PV interneurons in the “global” PV:LMO4 mice. To clarify, this reviewer does not require these experiments to be performed if the language is suitably changed throughout the manuscript in order to become more tempered and circumspect.

R> Thank you for this praise. **“tour de force!”**

Thank you for understanding that it is technically impossible at the current time to *only* ablate *Lmo4* in PV neurons at the AAV, since there is no AAV vector expressing CRE under a parvalbumin promoter to enable selective ablation of *Lmo4* in PV neurons at any selected brain region. We now acknowledge this limitation in our manuscript and have toned down our discussion of the LMO4/PTP1B in the ACC PV neurons and their relevance to behavior deficits.

2) A major point conveyed in this work is that the hyperexcitability of the PV-interneurons caused by LMO4 deletion interferes with proper cortical microcircuitry. The authors do a very good job looking at several electrophysiological properties of the circuit, assess local and long-range connectivity, but there is a clear lack of mechanistic insight into the cause of the hyperexcitability itself. The authors find indications that action potentials and resting membrane potential are changed in the PV neurons but only putatively indicate that this may be due to altered K⁺ currents while never directly addressing this point. It would be important to directly compare measures of K⁺ currents between genotypes that could allow for the observed differences and depart from conjecture.

R> To address this question, we have carried out additional experiments of I_A currents for all genotypes. We find no difference in the I_A currents in KO compared to WT and DKO mice (Sup Fig 3e). Thus, the main effect is likely through the leak conductance. Also, as requested by reviewer 3, we have provided zoomed-in insets for Fig. 2h, 2i, 7h and 7i to reveal the change in leak conductance and the delayed rectifier K⁺ currents.

3) The behavioral characterization performed in this work is cursory. There are, however, two possibilities that could easily improve the authors' claims and link to autism, by either looking at the ACC activity in the context of social behaviors, or by performing tests that are more ACC specific. This would be important particularly since the human genetics data linking LMO4 to autism is not very strong. For example, in the database pointed to by the authors there is a clearer connection to intellectual disability (6/9 cases) rather than autism (1/9 cases), so linking this circuit to the behavior analysis is critically important.

Are there behavioral tests that more specifically assess functions that are linked to ACC; or alternatively, happens to c-fos activation in the ACC of their animals following a social stimulus?

R> The ideal experiment would be to perform *in vivo* recording or imaging at the ACC while conducting the social interaction behavior tests. However, we do not have an *in vivo* recording system to conduct these experiments.

Of note, we now cite (page 2) a recent 2019 August report that appeared in *Nature Neuroscience*⁴: “An elegant recent study confirmed the relevance of the ACC to social interaction behaviors; selective ablation of the autism-risk gene *Shank3* in ACC pyramidal neurons caused social interaction deficits while restoration of *Shank3* only in ACC neurons rescued social deficits in global *Shank3* mutant mice⁴.”

As suggested by the reviewer, immunostaining of c-Fos revealed increased neuronal activity in L2/3 of the ACC 30 minutes after social interaction. These results are now presented in Fig. 1d and Fig. 7c. We found that compared to naïve mice, wild type mice experiencing social interaction showed an 8-fold increase in the number of c-Fos⁺ neurons. In contrast, only half as many c-Fos⁺ neurons were counted in PV-*Lmo4*KO mice after social interaction. Also, for PV-DKO mice, “Consistent with rescued behaviors, the number of c-Fos⁺ neurons after social interaction were not different between WT and PV-DKO mice (Fig. 7c)”. Also see below.

As for LMO4's link to autism, we agree that prior to our present study, there is limited support for this claim. We have toned down the statement relating to mutations that include LMO4.

4) While never directly intersecting with this work, the manuscript would be greatly improved if the authors discuss and contextualize their new finding in light of previous work showing a dysfunction in mGluR5 signaling in LMO4 KO mice. This would strengthen the discussion by offering other potential avenues that could help explain the animal's phenotype and some of the circuit alterations observed. The authors should also acknowledge recent evidences linking autism/intellectual disability to mGluR5 signaling as not to obviate this competing (but not mutually exclusive) possibility. In this sense, the following works should be cited.

<https://doi.org/10.1016/j.neuron.2015.02.015>

<https://doi.org/10.1038/mp.2015.113>⁵

<https://doi.org/10.1038/s41467-019-09382-9>

R> Thank you. We now include the following paragraph in our Discussion: “Our previous study reported that ablation of *Lmo4* in glutamatergic neurons (in *Camk2αCre/Lmo4*flox mice) hyperactivates PTP1B that dephosphorylates mGluR5 and impairs mGluR5-dependent endocannabinoid production⁶. mGluR5 function may be similarly affected in PV-*Lmo4*KO mice, due to PTP1B activation in PV neurons. Altered mGluR5 function has been associated with autism-like behaviors in mice lacking *Gprasp2*⁷. Moreover, PV neuron-specific ablation of mGluR5 induces compulsive repetitive behaviors and social novelty deficits associated with loss of PV neurons and reduced inhibitory currents (mIPSC) at the hippocampus⁵. In contrast, our PV-*Lmo4*KO mice display repetitive behaviors and social deficits without detectable PV neuron loss. Moreover, we detected increased spontaneous and evoked PV-mediated inhibitory currents in L2/3 pyramidal neurons of the ACC.”

5) Please provide a zoomed in of the optogenetic stimulus onset and postsynaptic response that reflect

mono versus di-synaptic response in the example in Figure 3 and 4.

R> Traces are now zoomed in and enlarged (new Fig 4b, 5b).

6) Murine genes should be indicated in italic

R> Corrected. Thank you.

7) The x-axis on figure 2d is cut

R> Corrected. Thank you.

8) In the main text, page 4, Supplementary figure 4b is being indicated, whereas 4c should be the one being noted.

R> Corrected (it is now Supplementary Fig. 4b,c on page 5). Thank you.

Reviewer #3 (Remarks to the Author):

The present manuscript reports alterations in FFI in the aCC following LMO4 and LMO4-PTP1B in PV neurons KO. The electrophysiological studies are well performed and present important data on the inhibitory circuit. The behavior is convincing but see comment below. IT is a well presented set of data. This study has to the body of knowledge regarding FFI and interneuron contribution to ASD phenotype. I nevertheless have the following set of comment:

R> Thank you.

Figure 2:

a. Please show some of the AP traces as well as whole cell currents.

R> Traces are now included (see Fig. 2c and Fig. 7e).

b. The IV curves in h show no significant difference. So it is unclear how this could account for the significant change in cd. In addition, there was almost no change at the resting membrane potential and thus it does not seem that the mentioned change in conductance accounts for the difference in resting membrane potential. Actually based on the IV curve, there is no difference around -60 or -65 mV so the cells should have the same resting membrane potential. Please clarify.

R> We apologize that our original displays were too small to show the differences. We now include zoomed-in insets for Fig 2h, 2i, 7h,7i. The magnified I-V curves reveal the differences between WT and KO.

c. Some of the data presented in supp Fig. 3 should be incorporated in the main figure. These data such as E/I ratio are important.

R> Yes, old Suppl. Fig. 3 is now new Fig. 3.

In the DKO, what is the expected effect of more KV current on cell firing and integration?

R> As we stated in the Results on page 10, 2nd paragraph, action potential width/duration, tied to Kv3.1 & Kv3.3, was restored to wild type levels in DKO mice (Supplemental Fig. 15). Yet, delayed rectifying KV current in DKO mice is exaggerated (over-compensated) and this likely sustains PV neuron firing at high frequency. KV currents in PV neurons modulate gamma oscillations in the somatosensory cortex⁸. Activation of PV neurons drives gamma oscillations and enhances signal transmission in neocortex by reducing circuit noise and amplifying circuit signals^{9, 10}. Whether exaggerated KV currents in DKO are sufficient to affect gamma oscillations *in vivo* in DKO mice remains to be determined. Impaired gamma oscillations at the mPFC leads to social interaction deficits in a mouse model of ASD¹¹. Given that social interactions are normalized in DKO mice, one would guess that gamma oscillations are likewise normal in these mice.

Figure 6c: there is a very large spread in the DKO. Did some of the mice present with locomotion issues?

R> There was no effect on locomotion (see suppl. Figs. 1 and 13).

1. Chung, W., *et al.* Social deficits in IRSp53 mutant mice improved by NMDAR and mGluR5 suppression. *Nat Neurosci* **18**, 435-443 (2015).
2. Wang, S., *et al.* Sh3rf2 Haploinsufficiency Leads to Unilateral Neuronal Development Deficits and Autistic-Like Behaviors in Mice. *Cell Rep* **25**, 2963-2971 e2966 (2018).
3. Moy, S.S., *et al.* Sociability and preference for social novelty in five inbred strains: an approach to assess autistic-like behavior in mice. *Genes Brain Behav* **3**, 287-302 (2004).
4. Guo, B., *et al.* Anterior cingulate cortex dysfunction underlies social deficits in Shank3 mutant mice. *Nat Neurosci* **22**, 1223-1234 (2019).
5. Barnes, S.A., *et al.* Disruption of mGluR5 in parvalbumin-positive interneurons induces core features of neurodevelopmental disorders. *Mol Psychiatry* (DOI:10.1038/mp.2015.113) (2015).
6. Qin, Z., *et al.* Chronic Stress Induces Anxiety via an Amygdalar Intracellular Cascade that Impairs Endocannabinoid Signaling. *Neuron* **85**, 1319-1331 (2015).
7. Edfawy, M., *et al.* Abnormal mGluR-mediated synaptic plasticity and autism-like behaviours in Gprasp2 mutant mice. *Nat Commun* **10**, 1431 (2019).
8. Joho, R.H., Ho, C.S. & Marks, G.A. Increased gamma- and decreased delta-oscillations in a mouse deficient for a potassium channel expressed in fast-spiking interneurons. *J Neurophysiol* **82**, 1855-1864 (1999).
9. Cardin, J.A., *et al.* Driving fast-spiking cells induces gamma rhythm and controls sensory responses. *Nature* **459**, 663-667 (2009).
10. Sohal, V.S., Zhang, F., Yizhar, O. & Deisseroth, K. Parvalbumin neurons and gamma rhythms enhance cortical circuit performance. *Nature* **459**, 698-702 (2009).

11. Cao, W., *et al.* Gamma Oscillation Dysfunction in mPFC Leads to Social Deficits in Neuroligin 3 R451C Knockin Mice. *Neuron* **97**, 1253-1260 e1257 (2018).

REVIEWERS' COMMENTS:

Reviewer #1 (Remarks to the Author):

The authors have fully addressed all of my comments. I do not have any further comments.

Reviewer #2 (Remarks to the Author):

The authors have address all my concerns. I endorse the acceptance of this manuscript for publication in its current form.

Reviewer #3 (Remarks to the Author):

answers are acceptable